# Poor adherence to TB diagnosis guidelines among under-five children with severe acute malnutrition in central India: A missed window of opportunity?

Akash Ranjan Singh[1], Amber Kumar[2], Hemant Deepak Shewade[3,4], Bhavna Dhingra[2]*

1 Government Medical College, Shahdol, India, 2 All India Institute of Medical Sciences (AIIMS), Bhopal, India, 3 The Union South East Asia, New Delhi, India, 4 International Union Against Tuberculosis and Lung Disease (The Union), Paris, France

* bhavna.pediatrics@aiimsbhopal.edu.in

**Data Availability Statement:** All relevant data are within the paper and its Supporting Information files. The data can also be accessed on following

## Abstract

### Background

In India, under-five children with Severe Acute Malnutrition (SAM) are referred to Nutritional Rehabilitation Centers (NRCs). NRCs screen the causes of SAM including tuberculosis (TB). The national TB programme recommends upfront testing with a rapid molecular test if TB is suspected in children.

### Objective

We estimated the yield of and adherence to the TB diagnostic guidelines (clinical assessment and assessment for microbiological confirmation) among under-five children with SAM admitted at NRCs (six in district Sagar and four in district Sheopur) of Madhya Pradesh, India in 2017. We also explored the challenges in screening from the health care providers' perspective.

### Methods

It was an explanatory mixed method study. The NRC records were reviewed This was followed by three key informant interviews and three focus group discussions among staff of NRC and TB programme. Manual descriptive thematic analysis was performed.

### Results

Of 3230, a total of 2665(83%) children underwent Mantoux test, 2438(75%) underwent physical examination, 2277(70%) were asked about the symptoms suggestive of TB, 1220 (38%) underwent chest radiograph and 485(15%) were asked for recent contact with TB. A total of 547(17%) underwent assessment for microbiological confirmation. Of 547, a total of 229 gastric aspirate specimens underwent rapid molecular test (24% positive) and 318 underwent sputum microscopy (44% positive). A total of 223 were diagnosed as TB (195 microbiologically and 28 clinically confirmed) and 209 were initiated on anti-TB treatment.

URL Quantitative data: https://www.dropbox.com/home/NRC%20paper/PloS%20resubmission_V2?preview=S2+Annex.xlsx Qualitative data: https://www.dropbox.com/home/NRC%20paper/PloS%20resubmission_V2?preview=S3+Annex.docx

**Funding:** This project was funded by National Health Mission, Government of Madhya Pradesh, India (Child Health-Nutrition/NHM/2017/25383).

**Competing interests:** The authors have declared that no competing interests exist.

The treatment outcome was favourable (cure or treatment completed) for 70(31%) and not recorded for 121(54%). The main perceived challenges in screening for TB were poor team skills, lack of diagnostic facilities and poor understanding of the guidelines due to inadequate training.

## Conclusion

Though NRCs provided a unique window of opportunity for the screening and management of TB among under-five children with SAM, the utilization of this opportunity remained suboptimal.

## Introduction

Globally in 2017, an estimated 10% of the total diagnosed tuberculosis (TB) patients were children (<15 years) but in India this was six percent among reported TB [1]. It is both under-diagnosed and under-reported. The major reasons include poor access to rapid diagnostic tests with high sensitivity and specificity, paucibacillary nature of childhood TB, and difficulty in obtaining quality specimen for microbiological confirmation. Hence, the majority of childhood TB is clinically diagnosed [2–6].

TB in children is an inevitable consequence of its presence in any community [7]. Childhood TB is an indicator of recent and ongoing transmission of *Mycobacterium tuberculosis* [8]. Thirty to forty percent of childhood TB is extrapulmonary and can present in a wide variety of anatomical sites [9, 10]. In infants, the time between infection and disease can be shorter than in older children, and the presentation may be more severe [11].

In India, among under-five children, 38% are stunted, 21% are wasted (a sign of acute undernutrition), while 36% are underweight [12, 13]. The Integrated Child Development Scheme of India guidelines identifies Severe Acute Malnutrition (SAM) as weight-for-height/length Z- score below 3SD of the median World Health Organization (WHO) child growth standards or a mid-upper arm circumference (MUAC) <115 mm or the presence of nutritional edema [13, 14]. SAM can be an indirect cause of child death by increasing the case fatality rate in children suffering from common illnesses including TB [13, 15]. As per WHO guideline for facility based management of children with SAM, they should be referred to Nutritional Rehabilitation Centres (NRCs at sub-district and district level). NRCs screen the causes of SAM including infectious causes like TB [13, 16].

Globally, studies have reported that 4–20% of children with SAM have active TB [17–19]. In India, studies form NRCs have reported a variable prevalence of TB among children with SAM, ranging from 22% to as low as <1% [20–25]. This could be due to the extent to which TB screening is implemented and the adherence to the diagnostic algorithm and this has not been assessed in India [25]. In 2016, India's national TB programme introduced a new pediatric TB diagnostic algorithm. This included up-front testing of gastric aspirate with a rapid molecular test if TB was suspected among children [26].

In order to understand the TB diagnosis gaps in children with SAM, our study aimed to determine the yield of and adherence to the TB diagnostic guidelines (clinical assessment and assessment for microbiological confirmation) among under-five children with SAM admitted at NRCs of Madhya Pradesh, India in 2017. We also explored the challenges in screening from the health care providers' perspective.

## Methods

### Study design

An explanatory type of mixed methods study design was used [27]. The descriptive quantitative phase involving secondary data collection was followed by a descriptive qualitative phase.

### Study setting

**General setting.** Madhya Pradesh is a state in central India and has one of the largest networks of community and facility-based centers for the management of malnutrition. In 2017, the state had 8,465 Anganwadi centres (AWCs), 12,670 mini-AWCs, 315 operational NRCs and two severe malnutrition treatment units (SMTUs). In 2017, a total of 70,859 SAM children were admitted at the NRCs and SMTUs [16]. The guidelines for the screening, diagnosis and treatment of TB are same at NRCs & SMTUs so SMTUs are described as any other NRC in the study.

The study was conducted in two districts: Sagar and Sheopur. Sagar district had six and Sheopur has four operational NRCs. As per operational guideline for a 10 bedded NRC, the human resource required is one medical officer, four nursing staff, one nutritional counsellor, one cook cum care taker, one attendant or cleaner and one medical social worker [28]. Most of the NRCs have three permanent staff; these are feeding demonstrator, staff nurse and a cook. The in-charge of these NRCs is preferably a Pediatrician or a medical Officer as per their availability in the health facility where the NRC is housed.

**Care pathway for identification of TB cases among children admitted at NRCs.**
Under-five children with SAM along with medical complications get referred to NRC by a community health worker or may come directly (self-referral). This child is screened for medical complications and causes of SAM [13]. TB might be one of the possible etiology or complication of SAM [29]. Hence, initial clinical assessment of every child for TB is done using these five criteria: i) history suggestive to TB; ii) history of contact with sputum smear positive TB in the family; iii) physical examination for the signs of TB disease; iv) chest radiograph and v) Mantoux test. The national TB programme recommends that children with presumptive TB should be screened by a rapid molecular test (cartridge-based nucleic acid amplification test (CBNAAT) also known as Xpert MTB/RIF©) for confirmation of TB [30]. There must be every effort to demonstrate *Mycobacterium tuberculosis*, in any of the biological specimens of the patient [31]. Hence, the gastric lavage (preferably) or induced sputum should be obtained for every admitted child in the NRC. In case, microbiological confirmation of TB cannot be established by this method, the in-charge pediatrician or medical officer can also make a clinical diagnosis depending on the initial assessment and further investigations. If the child is diagnosed as TB disease (either microbiologically or clinically) they get registered and directly observed treatment (DOT) is commenced as per their weight band.

### Study population

In the quantitative phase, under-five children with SAM, who were admitted to the NRCs of district Sagar and Sheopur between 1 January and 31 December 2017, were included. The selection of district Sagar and Sheopur was purely purposive. This was followed by qualitative phase in district Sagar. Concerned staff members were not available in Sheopur district because of strike during the qualitative study period (June 2018) and limited funds were available under the project, hence only one district was selected. For this, the representatives of the stakeholders involved in the care pathway of TB screening were selected if they were vocal, knowledgeable, and ready to participate in interviews. Selection was done to ensure maximum heterogeneity in the responses.

## Data collection and sources of data

In the quantitative phase, between February and May 2018, information related to screening, diagnosis and treatment was extracted and captured in a mobile based data collection tool [32]. The source was records of NRCs (admission register, case records, NRC SAM register), laboratory register of designated microscopic centers (DMC) and treatment cards of the children. In case of ambiguity/unavailability/unclear information, it was clarified with the staff present over the facility. Results of quantitative phase fed into the development of interview guide for the qualitative phase (see **S1 Annex** for interview guide).

In the qualitative phase (June 2018), we conducted three focus group discussions (FGDs), one with NRC staff (feeding demonstrators/auxiliary nurse midwife posted at NRC, n = 7), second with staff of DMCs (TB laboratory technician and TB health visitor, n = 7) and third with senior treatment supervisors (n = 6) of the district. The representation of every NRC and DMC of the district was ensured while inviting the participants for FGDs. For key informant interviews, the stakeholders who were involved in the care pathway were identified: pediatrician-in-charge of NRC (n = 1), staff nurse of the NRC (n = 1) and district TB officer (n = 1).

The qualitative enquiry was conducted in Hindi at the date and time convenient to participants. Two investigators carried out all the qualitative interviews, among them one was trained in conducting the qualitative study. Before the start of interview, participants were informed about the purpose of the study. An interview guide with broad open-ended questions was prepared, pilot-tested and used to conduct these interviews/discussions. Audio recording (after consent) and verbatim notes were taken. The time duration of key informant interviews ranged from 32 to 55 minutes and the FGDs ranged from 42 to 65 minutes. Translation and transcription (in English) was done within a week by the investigators. Field notes of observations during visits/interviews were also made.

## Data analysis

In the quantitative phase, data was extracted in Microsoft Excel 2010 (Microsoft, Redmond, WA, USA) and analyzed using STATA (version 12.1 STATA Corp., College Station, TX, USA). Key analytic outputs were the number (proportion) of children that were clinically assessed (each criteria presented separately) and the number (proportion) that underwent smear microscopy or rapid molecular tests to establish microbiological confirmation. It was operationally decided to include sputum microscopy for the assessment of microbiological confirmation. The STROBE guideline was followed for reporting the quantitative component of the study [33].

In the qualitative phase, transcripts obtained were compiled and discussed among investigators to become familiar with the data. Manual descriptive thematic analysis was used [34]. Codes and themes were reviewed to reduce bias and interpretive credibility. The decision on coding rules and theme generation was done by using standard procedures and with consensus among investigators. Both inductive and deductive codes were generated. Similar codes were combined into themes. To ensure that the results as true reflection of data, the codes/themes were reverted back to the original data. The themes were described and de-identified as representative statements were included in the results to illustrate the themes. The findings were reported by using COnsolidated criteria for REporting Qualitative research (COREQ) [35].

**Ethics.** The study was approved by Institutional Human Ethics Committee of the All India Institute of Medical Sciences, Bhopal, India (IHEC-LOP/2018/EF0076, dated 02/02/2018). Administrative approval was also obtained before starting the study. As the quantitative phase was based on record review, waiver for written informed consent was requested and

approved by the ethics committee. For the qualitative phase, written informed consent was taken (separately for audio recording) and the process was approved by the ethics committee. The data pertaining to the study participants (quantitative as well as qualitative) were fully anonymized.

## Results

### Quantitative phase

Of 3230 under-five children with SAM admitted at NRCs, 1552 (48%) were in Sagar and 1678 (52%) in Sheopur. Of 3230 children, 2263(70%) were identified by the Integrated Child Development Scheme services, 204 (6%) got admitted through healthcare delivery system, 85(3%) were self-referred and the source of referral was not recorded for 678 (21%) children (**Fig 1**).

As part of clinical assessment, 2665(83%) children underwent Mantoux test, of them 176 (6%) had a positive test; 2438(75%) underwent physical examination, of them one was suggestive of TB; 2277(70%) were asked about the symptoms suggestive of TB, of them 202 (6%) had symptoms suggestive of TB; 1220(38%) underwent chest radiograph, of them 159 (5%) had features suggestive of pulmonary TB; and 485(15%) were asked for recent contact with TB, of them 12 (0.4%) reported recent contact. (**Table 1**)

Gastric aspirate was obtained from 229(7%) children for CBNAAT and of them 55 (24%) were positive. Gastric aspirate specimen was obtained from 318(10%) for smear microscopy and of them 140 (44%) were positive. It was also revealed that in certain cases, where specimen was being sent for CBNAAT, smear microscopy was not being done. Of 3230, a total of 547 (17%) were tested either by smear microscopy or CBNAAT. A total of 223 (7%) were diagnosed as TB: 195 (6%) microbiologically confirmed and 28 (1%) clinically confirmed. (**Fig 1**)

The proportion being assessed for microbiological confirmation and the proportion diagnosed as TB, stratified by NRCs in the two districts is depicted in **Table 2**. The extent of screening varied across NRCs in Sagar while it was consistently low in Sheopur.

Of the 223 children with TB, anti-TB treatment was initiated for 209 (94%). Of 209, the treatment outcome was favourable for 70 (34%) (cured or treatment completed). Treatment

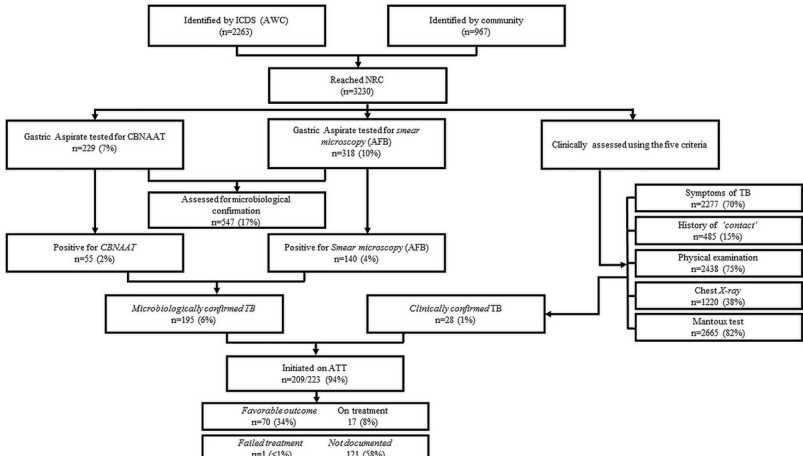

**Fig 1. Care pathway of TB management for under-five children with SAM admitted at Nutrition Rehabilitation Centers (NRC) of Sagar and Sheopur district, Madhya Pradesh, India (2017).** ICDS: Integrated Child Development Services; AWC; Anganwadi Center, SAM; Severely Malnourished children, AFB; Acid Fast Bacilli, NRC; Nutritional Rehabilitation Center, $; Others: Healthcare system/Self-referred/Not recorded, CBNAAT; Cartridge-Based Nucleic Acid Amplification Test, ATT; Antituberculosis treatment.

**Table 1. Socio-demographic profile and clinical assessment of under-five children with SAM admitted in Nutrition Rehabilitation Centers (NRC) of Sagar and Sheopur district, Madhya Pradesh, India (2017).**

| Variables | | Sagar (N = 1552) | Sheopur (N = 1678) | Total (N = 3230) |
|---|---|---|---|---|
| | | n(%)# | n(%)# | n(%)# |
| **Age (in years)** | | | | |
| | Up to 1 | 574(37) | 727 (43) | 1301 (40) |
| | 1 to 5 | 978(63) | 951 (57) | 1929 (60) |
| **Sex** | | | | |
| | Male | 740(48) | 829 (49) | 1569 (48) |
| | Female | 812(52) | 849 (51) | 1661 (52) |
| **Residence** | | | | |
| | Rural | 242(16) | 1 (<1) | 243 (7) |
| | Urban | 1 (0) | 1 (<1) | 2 (<1) |
| | Not recorded | 1,309(84) | 1676 (100) | 2985 (92) |
| **Socio-economic status of family** | | | | |
| | BPL | 13(1) | 283 (17) | 296 (9) |
| | APL | 1 (0) | 2 (0) | 3 (<1) |
| | Not recorded | 1538(99) | 1393 (83) | 2931 (91) |
| **History of admission to NRC in past** | | | | |
| | Yes | 0 (0) | 4 (<1) | 4 (<1) |
| | No | 1549(100) | 1671 (100) | 3220 (99) |
| | Not recorded | 3(<1) | 3 (<1) | 6 (<1) |
| **History suggestive of TB** | | | | |
| | Positive/Asked | 202/604(33) | 0/1673(0) | 202/2277 (9) |
| **History of contact to TB case** | | | | |
| | Positive/Asked | 12/388 (3) | 0/97(0) | 12/485 (2) |
| **Physical examination** | | | | |
| | Positive/Performed | 1/795 (<1) | 0/1643(0) | 1/2438 (<1) |
| **Chest X ray** | | | | |
| | Positive/Performed | 140/697(20) | 19/523(4) | 159/1220 (13) |
| **Mantoux test** | | | | |
| | Positive/Performed | 146/1172(13) | 30/1493(2) | 176/2665 (7) |

# column percentage, TB: Tuberculosis, BPL: Below Poverty Line; APL: Above Poverty Line, NRC: Nutritional Rehabilitation Center, DOT: directly observed treatment.

was still going on for 17(8%), outcome was not recorded for 121(58%) and one child failed the treatment (**Fig 1**).

## Qualitative phase

Qualitatively, the barriers for poor screening of under-five children with SAM at NRCs were explored and five themes were identified (**Fig 2**).

**Poor record keeping.** Among the children reaching the NRCs, the source of referral was not documented in one-fourth of cases. Even for children referred to higher facilities from NRCs, this information was poorly recorded. The NRC staff did not perceive this as an important job responsibility. (Field notes)

**Lack of training.** There was lack of training of peripheral health workers and if they had received training, most did not remember the training content. The NRC staff (feeding demonstrator) stated during an FGD.

**Table 2. Under-five children with SAM admitted, assessed for microbiological confirmation and diagnosed as TB at Nutrition Rehabilitation Centers (NRC) of Sagar and Sheopur district, Madhya Pradesh, India (2017).**

| Name of NRC | Available beds | Total admitted | Assessed for microbiological confirmation | Diagnosed as TB [#] |
|---|---|---|---|---|
| | | N | n (%) | n (%) |
| **Total** | | **3230** | **547 (17)** | **223 (7)** |
| **Sagar district** | | **1552** | **481 (31)** | **174 (11)** |
| District hospital | 20 | 557 | 196 (35) | 77(14) |
| Bina | 10 | 205 | 75 (37) | 73(36) |
| Khurai | 10 | 213 | 22(10) | 18(8) |
| Deori | 10 | 186 | 11(6) | 6(3) |
| Malthon | 10 | 190 | 176(93) | 0(0) |
| Godhakota | 10 | 201 | 1(<1) | 0(0) |
| **Sheopur district** | | **1678** | **66(4)** | **49 (3)** |
| District hospital | 10 | 610 | 46 (8) | 30 (5) |
| Karahal | 10 | 491 | 2 (<1) | 1 (<1) |
| Vijaypur | 10 | 341 | 15 (4) | 15 (4) |
| Baroda | 10 | 236 | 3 (1) | 3 (1) |

Row percentage maintaining 'Admitted' as the denominator.

TB–tuberculosis.

[*]Underwent smear microscopy or rapid molecular test for microbiological confirmation; [#] includes microbiologically as well as clinically confirmed TB.

No, we have not gone through any training pertaining to screening/diagnosis of TB at NRCs

**Reluctance for invasive procedure.** NRC staff perceived that obtaining biological specimen like gastric aspirate from children was a difficult task. One NRC staff (staff nurse) with more than five years' experience, when asked about the gastric aspirate, described her problem,

*No! Nobody in the hospital is trained to do it* [gastric aspirate]. *They have never been trained for this.*

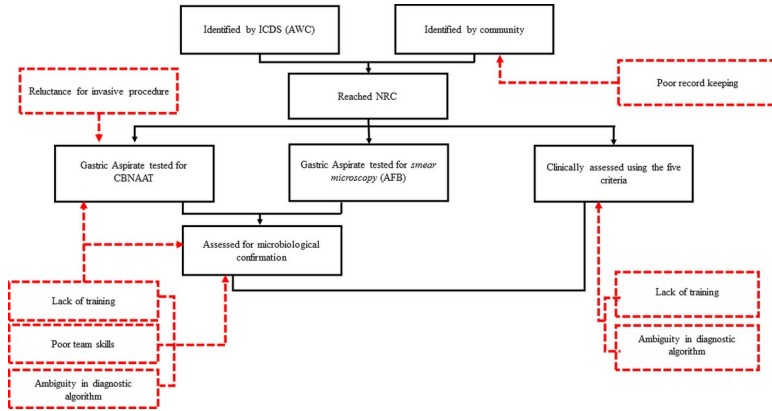

**Fig 2. Perceived challenges of healthcare workers in screening of TB among under-five children with SAM admitted at Nutrition Rehabilitation Centers (NRC) of Sagar and Sheopur district, Madhya Pradesh, India (2017).** ICDS: Integrated Child Development Services; AWC; Anganwadi Center, Sputum smear microscopy, AFP; Acid Fast Bacilli, NRC; Nutritional Rehabilitation Center, CBNAAT; Cartridge-Based Nucleic Acid Amplification Test, ATT; Antituberculosis treatment.

**Poor team skills.** Poor team skills were observed in the form of lack of motivations of the in-charge pediatricians. This could be due to poor understanding of guidelines, lack of diagnostic facilities or non-feasibility of diagnostic tests. It eventfully led to poor adherence to the diagnostic guidelines.

*The situation is different at different NRCs. We do not have facility for chest radiograph, CBNAAT and better guidance is needed to facilitate all the diagnostic tests. Considering all this the mainstay of diagnosis of TB is the Mantoux test barring at the NRC at district hospital.* (DMC staff)

An NRC in-charge felt that lack of collection of gastric aspirate of CBNAAT was due to the poor motivation of staff nurse.

*We used to write* [investigation request] *for gastric aspirate, but our staff nurse cannot do this. The challenge here is to motivate them for gastric Aspirate.* (Pediatrician of an NRC)

**Ambiguity in diagnostic algorithm.** CBNAAT was not available in all facilities. The screening and diagnostic guidelines were not explicit regarding what was to be done when certain tests/facilities were either not available or not feasible due to lack of assured specimen transport; therefore the screening guidelines were not consistently followed across the NRCs.

*Children admitted at Sagar NRC undergo for CBNAAT, but for other places, it mainly depends on the wisdom of the doctor and feasibility of the tests. In most places, the availability of pediatrician is not confirmed and it is difficult for other staff members to diagnose or decide.* (FGD, DMC staff)

*At Sagar NRC, we use to send gastric aspirate of almost every child for CBNAAT since last few months. But for other NRCs, the decision of treatment is solely based on the result of Mantoux test, as no other tests are available.* (DMC staff)

## Discussion

### Key findings

NRCs provide a unique window of opportunity for the screening of TB among under-five children with SAM, the utilization of this opportunity remained suboptimal. In the context of poor implementation of TB screening, the yield of smear microscopy and CBNAAT in the small subgroup of children who underwent testing cannot be extrapolated to all the children admitted at NRC (n = 3230). The diagnostic guidelines implemented at different NRCs were different. According to our findings, adherence to screening for TB was low because of poor team skills which may be due to busy schedule of in-charge pediatrician or their involvement in other programmes; non-availability of diagnostic facilities and poor knowledge of these guidelines. Furthermore, the treatment outcomes of more than half were not available. Because of poor adherence to diagnostic guidelines and suboptimal initial assessment, we are missing a lot of TB among under-five children with SAM.

Poor record keeping was consistently seen. Information related to source of referral of children to NRC, tests advised and date of testing was not consistently recorded. It appeared that there was a poor understanding of the denominator and a lack of cohort-wise tracking. High positivity rates among those tested indicates that those who were tested were retrospectively entered in the records.

During the initial assessment, less than three-fourth children were asked about the history suggestive of TB and history of contact to TB case was investigated in less than one-fourth children. Despite being simple to evaluate and hardly time-consuming, they were missed. The staff either undermined the importance or did not record the details in the case sheets. Similarly, few children had a record of physical examination. Only one child was found with signs suggestive of TB. This is surprising considering tubercular lymphadenitis is most common manifestation of extrapulmonary TB in children [7]. Studies from central India reported 0.4 to 22% prevalence of tubercular lymphadenitis among children (age less than 15 years) [36, 37]. Despite the fact that these studies did not address the particular age group of under-five children but still the proportion is very low in our study especially among children with SAM. Only about one-third children underwent chest radiograph and three-fourth underwent Mantoux test. This affects clinical diagnosis. It might be due to poor team skills that might have led to erratic usage of these diagnostics and poor coordination of NRC staff with concerned departments.

Although both gastric lavage and induced sputum are complementary to each other with comparative yield, it was interesting to note that gastric aspirate was preferred [38]. Induced sputum is cumbersome and more time-consuming process than gastric aspirate. Therefore, it is difficult to perform induced sputum at NRC attached health facilities at a regular basis.

A similar study carried out in 2012 in six sub-district level administrative units of Karnataka (a state in south India) reported bottlenecks in the diagnosis of TB at NRC. These were non-availability of full-time pediatrician, non-availability of diagnostic facilities and poor adherence to the diagnostic algorithm because of its non-feasibility [22]. This is similar to the findings of our study. The studies from other high burden countries also suggest the screening of childhood TB is always difficult and mostly delayed [39–46].

## Recommendations

The following are our recommendations i) focus on reforming and reinforcement of result oriented quality training methods of NRC staff ii) ensure sustained availability and functionality of diagnostic equipment required for the screening iii) providing an assured specimen transport mechanism iv) standardized and simplified mechanism of record keeping and v) close integration between NRC and TB programme for treatment adherence during follow up visits at NRC.

Various studies have suggested capacity building of TB programme staff can significantly improve the TB outcomes [47–51]. Despite increase in expenditure towards training programmes in middle and low income countries, its robust evaluation is needed [52]. Ensuring sustained availability of diagnostic equipment will definitely improve the screening and diagnosis process of TB [40–44]. There are obvious reasons to believe that good record keeping process and close integration between NRC and TB programme will yield better results.

## Strengths and limitations

The major strength of this study was its mixed methods design which proved invaluable in complementing and supplementing quantitative and qualitative findings. However, few limitations of the study are worth noting. First, the selection of two districts restricts the generalization of the findings of the quantitative phase. Second, the qualitative phase was undertaken only in one district. Qualitative systematic enquiry did not yield details about TB treatment adherence and high proportion of children with non-evaluated TB treatment outcomes.

## Conclusion

To conclude, present study documented poor adherence to initial assessment and TB diagnostic guidelines among under-five children with SAM admitted at NRCs of two districts in central India. The main challenges perceived by the healthcare providers are poor team skills and poor understanding of relevant guidelines due to ambiguity in the diagnostic algorithm. Study recommends quality training to NRC staff, sustained availability of diagnostic equipment, uniform and simplified mechanism of record keeping and assured specimen transport mechanism.

## Supporting information

**S1 Annex.**
(DOCX)

**S2 Annex.**
(XLSX)

**S3 Annex.**
(DOCX)

## Acknowledgments

The authors thank the National TB Elimination Programme and National Health Mission of Madhya Pradesh (India) for providing support and necessary guidance. We express our gratitude for our data collection team, comprising of Mrs. Seema Gotiwale (Project Coordinator), Mr. Saurabh Singh and Mr. Saurabh Rajak (both Field Data Collector).

   **Disclaimer:** The contents of this paper do not necessarily reflect the views of the institutions the authors are affiliated to.

## Author Contributions

**Conceptualization:** Akash Ranjan Singh, Amber Kumar, Bhavna Dhingra.

**Data curation:** Akash Ranjan Singh, Bhavna Dhingra.

**Formal analysis:** Akash Ranjan Singh, Hemant Deepak Shewade, Bhavna Dhingra.

**Funding acquisition:** Bhavna Dhingra.

**Investigation:** Akash Ranjan Singh, Amber Kumar, Bhavna Dhingra.

**Methodology:** Akash Ranjan Singh, Amber Kumar, Bhavna Dhingra.

**Project administration:** Akash Ranjan Singh, Amber Kumar, Bhavna Dhingra.

**Resources:** Akash Ranjan Singh, Amber Kumar, Bhavna Dhingra.

**Software:** Akash Ranjan Singh.

**Supervision:** Akash Ranjan Singh, Amber Kumar, Bhavna Dhingra.

**Validation:** Akash Ranjan Singh, Amber Kumar, Bhavna Dhingra.

**Visualization:** Akash Ranjan Singh, Amber Kumar, Bhavna Dhingra.

**Writing – original draft:** Akash Ranjan Singh.

**Writing – review & editing:** Akash Ranjan Singh, Amber Kumar, Hemant Deepak Shewade, Bhavna Dhingra.

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
