## [Decision Letter · Decision Letter 0]

6 Nov 2020

PONE-D-20-30406

Screening of childhood TB at the nutritional rehabilitation centers of Madhya Pradesh: a missed window of opportunity?

PLOS ONE

Dear Dr. Dhingra,

Thank you for submitting your manuscript to PLOS ONE. After careful consideration, we feel that it has merit but does not fully meet PLOS ONE’s publication criteria as it currently stands. Therefore, we invite you to submit a revised version of the manuscript that addresses the points raised during the review process.

Please submit your revised manuscript. If you will need more time than this to complete your revisions, please reply to this message or contact the journal office at plosone@plos.org. Please include the following items when submitting your revised manuscript:

We look forward to receiving your revised manuscript.

Kind regards,

Frederick Quinn

Academic Editor

PLOS ONE

Journal Requirements:

3. When reporting the results of qualitative research, we suggest consulting the COREQ guidelines: http://intqhc.oxfordjournals.org/content/19/6/349. In this case, please consider including more information on the following: 1) the number of interviewers, their training and characteristics, 2) please provide the interview guide used 3)  additional information regarding the number of participants in each focus group and 4) nformation regarding the pilot of the guides and whether any modifications were made.

4. In ethics statement in the manuscript and in the online submission form, please provide additional information about the patient records/samples used the qualitative aspect of the study. Specifically, please ensure that you have discussed whether all data/samples were fully anonymized before you accessed them.

5. Please include your tables as part of your main manuscript and remove the individual files. Please note that supplementary tables (should remain/ be uploaded) as separate "supporting information" files

6. We note you have included a table to which you do not refer in the text of your manuscript. Please ensure that you refer to Table 1 and 2 in your text; if accepted, production will need this reference to link the reader to the Table.

7.We note that [Figure(s) 1] in your submission contain map images which may be copyrighted. All PLOS content is published under the Creative Commons Attribution License (CC BY 4.0), which means that the manuscript, images, and Supporting Information files will be freely available online, and any third party is permitted to access, download, copy, distribute, and use these materials in any way, even commercially, with proper attribution. For these reasons, we cannot publish previously copyrighted maps or satellite images created using proprietary data, such as Google software (Google Maps, Street View, and Earth). For more information, see our copyright guidelines: http://journals.plos.org/plosone/s/licenses-and-copyright.

1.    You may seek permission from the original copyright holder of Figure(s) [1] to publish the content specifically under the CC BY 4.0 license. 

8. Please ensure that you refer to Figure 2 in your text as, if accepted, production will need this reference to link the reader to the figure.

9. PLOS requires an ORCID iD for the corresponding author in Editorial Manager on papers submitted after December 6th, 2016. Please ensure that you have an ORCID iD and that it is validated in Editorial Manager. To do this, go to ‘Update my Information’ (in the upper left-hand corner of the main menu), and click on the Fetch/Validate link next to the ORCID field. This will take you to the ORCID site and allow you to create a new iD or authenticate a pre-existing iD in Editorial Manager. Please see the following video for instructions on linking an ORCID iD to your Editorial Manager account: https://www.youtube.com/watch?v=_xcclfuvtxQ

10. Thank you for stating the following in your Competing Interests section: 

[None].

Reviewers' comments:

Reviewer's Responses to Questions

**Comments to the Author**

1. Is the manuscript technically sound, and do the data support the conclusions?

Reviewer #1: Yes

Reviewer #2: Yes

2. Has the statistical analysis been performed appropriately and rigorously? 

Reviewer #1: Yes

Reviewer #2: Yes

3. Have the authors made all data underlying the findings in their manuscript fully available?

Reviewer #1: Yes

Reviewer #2: Yes

4. Is the manuscript presented in an intelligible fashion and written in standard English?

Reviewer #1: Yes

Reviewer #2: Yes

5. Review Comments to the Author

Reviewer #1: PONE-D-20-30406_reviewer comments

Needs major revisions

• Overall: This is an important study to show missed opportunities of screening, diagnosing, and treating TB in children admitted for SAM in India. Please see below for specific recommendations for improvement. Additionally, my main suggestion would be to expand the scope of the paper (title, etc) to include not only TB screening, but also TB diagnosis and treatment in your cohort. This is covered in your results nicely, and would make the paper much stronger and comprehensive if it focuses on TB diagnosis and TB treatment initiation (in addition to TB screening).

• Cover page:

o Please correct the title. It appears that “To, The Editor the full title…” was accidentally copy and pasted. Please only list the full title.

o Key words: also include “TB screening” as a key word

• Title:

o Recommend including the main findings/message in the title as well. How was it a missed opportunity? Let your read know the main findings/message of the TB screening in the Title.

• Abstract:

o Please fix various typographical and grammar errors in the abstract (e.g. extra words, spaces, unnecessary quotation marks, etc)

o In Methods, include the time period of the study (dates of review)

o Results: in addition to # screened, please also include # diagnosed with TB, # treated for TB, and # finishing treatment/cured (and # deaths/LTFU) of those 3230 admitted. Also please include results of the gastric aspirates mentioned in the Background (how many had gastric aspirates, how many had positive results….as well as any other TB diagnostic test among those 3230).

• Introduction:

o Lines 78-79, please make it clearer that the % of reported TB in children was only 6%. As written it is not very clear if that is the intended message. Also, you can delete the “There are enough reasons…” sentence, as your next sentence is sufficient (if you slightly modify it).

o Lines 102-104. Other than gastric aspirate, are there any other recommended TB screening for children with SAM? For example, TB screening questions, score chart, CXR, TST/IGRA, etc? Please mention the entire recommended TB screening protocol for children with TB….or if it is only gastric aspirate, please mention that specifically.

• Methods

o Lines 117-120: when listing numbers, either use a comma for thousands (e.g. 8,465) or remove the space between the thousands and hundreds place (8465). Having a space (“8 465”) makes it more difficult to read.

o General settings section: you mention both SMTUs and NRCs…but then your paper only focuses on NRCs? Need more clarification between the two, and why only NRCs examined in the study (what about the SMTUs??)

o Line 149: Please clarify the exact dates of 2017 reviewed (e.g. 1st January to 31st December)

o Lines 151-153: more details needed on how health care workers were selected for interviews. You need to show your reader that you took steps and precautions to avoid and minimize selection bias and other biases. More details needed to clearly how you ensured “maximum heterogeneity in the responses.”

o Data analysis section: the main concern is that your paper’s title focuses on “TB Screening” (which you explained the five steps of doing that in the methods), but then your main output is only microbiologic confirmation. These are two different things….screening vs confirmation of disease. Thus, your main output is not actually a screening output (if screening was done or not in the children). Rather, your main output is microbiological confirmation of TB disease in these children. Very different things. And as you mention in your Introduction, confirmation of TB in young children (especially when SAM) is very difficult and rare.

• Results

o Qualitative phase section: please include results of all of the screening techniques you mention (i.e. of those screened for XYZ, how many screened positive). Of the Mantoux test, physical exam, suggestive symptoms, CXR, history of contact….we need to know how many screened positive for each of those.

Also, please then list how many of the positive screens (and how many of the children with SAM) were diagnosed with TB, and how many started treatment.

o Line 225: unclear what “infective training” refers to. Need more specifics. Lack of confidence on TB symptoms, reading CXR, score charts, Mantoux testing, sputa extraction? Need more details on the gap.

o Line 230: is there also reluctance to sputum induction? Or is one method preferred over the other (which is very important information to share!!). which method is accepted more?

• Discussion

o Lines 282-286. For these five recommendations, can you offer any background or evidence base (citations) that these are effective strategies for your setting? They all seem reasonable, but would strengthen the paper if you can show such interventions have been effective at other settings in India, or globally.

o Lines 294-297: you can remove these from discussion. These were already covered in Methods and Results. Focus instead on main messages of the study (main take away points)

o Line 300-301: you mention “availability of diagnostic equipment” here…but in results, this was not mentioned and not listed as a challenge from the interviews. Therefore, can not mention it here unless it is referenced above in results.

• Figures/Tables

o Figures 2 and 3 are excellent, and show the entire TB treatment cascade. I would focus more on highlighting these tables in the text, and focusing on the “leakage” and issues at each step!

o Again, your title/paper focuses mostly on TB screening…but you have excellent data along the screening, diagnosis, treatment steps. I would highly recommend you expand the scope of title and paper to include these as well (and not limit it to just screening)

o Table 1

Clarify that “More than 1” actually is only “1-5 years old”

Residence: remove this as 84-99% didn’t have it recorded (thus not useful)

• Same for SES of family (if not recorded, not useful to include)

In Table 1, can you also include TB screening (the 5 items) to show how many of the children had each of those 5 screens, and then how many had positive screens of those 5

o Table 2: please also include the % in ( ) after each number. The first three columns are missing the (%) and only list N

o Both tables: put your abbreviations footnote below the table (separate) and not as a final row within the table.

Reviewer #2: Comments to the Author:

Point 1: In the section of Introduction (Page 10, Line 2), the “6.%” was incomplete information. The author should check it and ensure the accuracy of the information.

Point 2: In the section of Data collection and sources of data (Page 13, Line 16), the full name of “DMCs” should be given when it first appeared in the text.

Point 3: In the section of Result, the format of the number of cases and percentages in Figure 2 should be unified. Besides, some footnotes were inconsistent with the contents in the Figure2 and Figure3.

Point 4: Detailed descriptions of tables 1 and 2 cannot be found in the Result section, please check it carefully.

Point 5: The formats of the statistical tables were not standard, and the contents of footnotes were inconsistent with the contents of the tables.

Point 6: The corresponding percentages were missing after number in Table 2.

Point 7: There were too much description of results in the discussion section. The author should refer to other studies for further discussion.

Point 8: There were many spelling errors in the manuscript. The author should check and correct it.

6. PLOS authors have the option to publish the peer review history of their article (what does this mean?). If published, this will include your full peer review and any attached files.

Reviewer #1: No

Reviewer #2: **Yes: **WangXueMei

---

## [Author Response · Author response to Decision Letter 0]

3 Dec 2020

RESPONSE TO COMMENTS

EDITORIAL STAFF COMMENTS

COMMENTS

RESPONSE 

Thanks for your comment. We tried to best of our capacity to meet the journal’s guideline. The revised manuscript is as per journal requirements

COMMENTS

RESPONSE 

Thanks for your comment. We tried to meet the journal’s guideline related to Supporting documents. The revised manuscript is as per journal requirements

COMMENTS

3. When reporting the results of qualitative research, we suggest consulting the COREQ guidelines: http://intqhc.oxfordjournals.org/content/19/6/349. In this case, please consider including more information on the following: 1) the number of interviewers, their training and characteristics, 2) please provide the interview guide used 3) additional information regarding the number of participants in each focus group and 4) Information regarding the pilot of the guides and whether any modifications were made.

RESPONSE 

Thanks for your comment. We have incorporated the desired information in the manuscript. Please refer to lines 183-2045 of revised manuscript with track changes.

COMMENTS

4. In ethics statement in the manuscript and in the online submission form, please provide additional information about the patient records/samples used the qualitative aspect of the study. Specifically, please ensure that you have discussed whether all data/samples were fully anonymized before you accessed them.

RESPONSE 

Thanks for your comment. The data pertaining to the study participants (quantitative as well as qualitative) were fully anonymized. We have mentioned this in ethics statement and also in online submission form.

COMMENTS

5. Please include your tables as part of your main manuscript and remove the individual files. Please note that supplementary tables (should remain/ be uploaded) as separate "supporting information" files

RESPONSE 

Thanks for your suggestion. We have made this correction in the manuscript as per suggestion.

COMMENTS

6. We note you have included a table to which you do not refer in the text of your manuscript. Please ensure that you refer to Table 1 and 2 in your text; if accepted, production will need this reference to link the reader to the Table.

RESPONSE 

Thanks for pointing our mistake. We have made this correction in the manuscript as per suggestion. Table 1 and 2 have been cited in the text.

COMMENTS

7.We note that [Figure(s) 1] in your submission contain map images which may be copyrighted. All PLOS content is published under the Creative Commons Attribution License (CC BY 4.0), which means that the manuscript, images, and Supporting Information files will be freely available online, and any third party is permitted to access, download, copy, distribute, and use these materials in any way, even commercially, with proper attribution. For these reasons, we cannot publish previously copyrighted maps or satellite images created using proprietary data, such as Google software (Google Maps, Street View, and Earth). For more information, see our copyright guidelines: http://journals.plos.org/plosone/s/licenses-and-copyright.

1. You may seek permission from the original copyright holder of Figure(s) [1] to publish the content specifically under the CC BY 4.0 license. 

RESPONSE 

Thanks for suggestion. We have removed the figure 1 & renumbered the remaining ones.

COMMENTS

8. Please ensure that you refer to Figure 2 in your text as, if accepted, production will need this reference to link the reader to the figure.

RESPONSE 

Thanks for pointing our mistake. We have made this correction in the manuscript as per suggestion. 

COMMENTS

9. PLOS requires an ORCID iD for the corresponding author in Editorial Manager on papers submitted after December 6th, 2016. Please ensure that you have an ORCID iD and that it is validated in Editorial Manager. To do this, go to ‘Update my Information’ (in the upper left-hand corner of the main menu), and click on the Fetch/Validate link next to the ORCID field. This will take you to the ORCID site and allow you to create a new iD or authenticate a pre-existing iD in Editorial Manager. Please see the following video for instructions on linking an ORCID iD to your Editorial Manager account: https://www.youtube.com/watch?v=_xcclfuvtxQ

RESPONSE 

Thanks for your suggestion. We are providing the ORCID ID of the corresponding author.

COMMENTS

10. Thank you for stating the following in your Competing Interests section: 

[None].

RESPONSE

Thanks for pointing our mistake, we will completes our Competing Interests on the online submission as “The authors have declared that no competing interests exist."

REVIEWER COMMENTS 

REVIEWER 1

(Line numbers in our response are with reference to the revised manuscript with track changes)

COMMENT

Overall: This is an important study to show missed opportunities of screening, diagnosing, and treating TB in children admitted for SAM in India. Please see below for specific recommendations for improvement. Additionally, my main suggestion would be to expand the scope of the paper (title, etc) to include not only TB screening, but also TB diagnosis and treatment in your cohort. This is covered in your results nicely, and would make the paper much stronger and comprehensive if it focuses on TB diagnosis and TB treatment initiation (in addition to TB screening).

RESPONSE

We appreciate the reviewer for his words of encouragement. We have documented the level of initial assessment ,adherence to TB diagnostic guidelines, yield of microbiological confirmation among those undergoing testing. We also looked at treatment initiation and outcome (for the purpose of completing the care cascade). The qualitative part mostly focused on gaps in TB screening. Therefore we have limited our title to TB screening / diagnosis. To make the title more catchy (in line with you next comment), we have revised it as follows

“Poor adherence to TB diagnosis guidelines among under-five children with severe acute malnutrition in central India: a missed window of opportunity?”

COMMENT

Cover page:

Please correct the title. It appears that “To, The Editor the full title…” was accidentally copy and pasted. Please only list the full title. 

RESPONSE

Thanks for pointing out the mistake. We have corrected the title. 

COMMENT

Key words: also include “TB screening” as a key word

RESPONSE

Thanks for suggestion. We have made the change

COMMENT

Title: 

Recommend including the main findings/message in the title as well. How was it a missed opportunity? Let your read know the main findings/message of the TB screening in the Title.

RESPONSE

Thank you. We have incorporated this suggestion and included the main message / finding in the title. 

COMMENT

Abstract:

Please fix various typographical and grammar errors in the abstract (e.g. extra words, spaces, unnecessary quotation marks, etc)

RESPONSE

Thanks for suggestion. Two authors have carefully gone through the manuscript and corrected the typo and grammar errors.

COMMENT

In Methods, include the time period of the study (dates of review)

RESPONSE

Thanks for the suggestion. We have included the period of review (Feb-May 2018). Please refer to lines 78-79 and 185-86 of revised manuscript with track changes. 

COMMENT

Results: in addition to # screened, please also include # diagnosed with TB, # treated for TB, and # finishing treatment/cured (and # deaths/LTFU) of those 3230 admitted. Also please include results of the gastric aspirates mentioned in the Background (how many had gastric aspirates, how many had positive results….as well as any other TB diagnostic test among those 3230).

RESPONSE

 Thanks for suggestion. We have made the changes. (lines 82-85 of revised manuscript with track changes). We have reproduced the revised section below

“Of 3230 admitted children with SAM, 547(17%) were screened for TB as per guidelines, 223(7%) were diagnosed and 209 (6%) were treated for TB. Of 209 on treatment, outcome was documented for 87 (4%). Among 318(10%) whose gastric aspirate was obtained, 140 (4.3%) were microbiologically confirmed.”

COMMENT

Introduction:

Lines 78-79, please make it clearer that the % of reported TB in children was only 6%. As written it is not very clear if that is the intended message. Also, you can delete the “There are enough reasons…” sentence, as your next sentence is sufficient (if you slightly modify it).

RESPONSE

Thanks for suggestion. We have made the changes. (lines 93-94 of revised manuscript with track changes)

COMMENT

Lines 102-104. Other than gastric aspirate, are there any other recommended TB screening for children with SAM? For example, TB screening questions, score chart, CXR, TST/IGRA, etc? Please mention the entire recommended TB screening protocol for children with TB….or if it is only gastric aspirate, please mention that specifically. 

RESPONSE

Thanks for suggestion. , the gastric lavage (preferably) or induced sputum should be obtained for every admitted child in the NRC for the CBNAAT depending on feasibility. In case, the diagnosis of TB cannot be established by this method, the in-charge pediatrician or Medical Officer can also make a clinical diagnosis depending on the initial assessment and further investigations. The screening, diagnosis and treatment of TB for the children admitted at NRC is described in detail under the following section under settings‘’ Care pathway for identification of TB cases among children admitted at NRCs’ (lines 153-171 of revised manuscript with track changes)

COMMENT

Methods

Lines 117-120: when listing numbers, either use a comma for thousands (e.g. 8,465) or remove the space between the thousands and hundreds place (8465). Having a space (“8 465”) makes it more difficult to read. 

RESPONSE

Thanks for suggestion. We have made the change as suggested.

COMMENT

General settings section: you mention both SMTUs and NRCs…but then your paper only focuses on NRCs? Need more clarification between the two, and why only NRCs examined in the study (what about the SMTUs??)

RESPONSE

Thanks for pointing this out. The guidelines for the screening, diagnosis and treatment of TB is same at NRC and SMTUs, so for the simplicity of the readers we described the SMTUs as any other NRC in the paper. (lines 144-45 of revised manuscript with track changes) 

COMMENT

Line 149: Please clarify the exact dates of 2017 reviewed (e.g. 1st January to 31st December)

RESPONSE

Thanks for pointing this out. We have now clarified this. (line 175 of revised manuscript with track changes)

COMMENT

Lines 151-153: more details needed on how health care workers were selected for interviews. You need to show your reader that you took steps and precautions to avoid and minimize selection bias and other biases. More details needed to clearly how you ensured “maximum heterogeneity in the responses.”

RESPONSE

Thanks for the suggestion. The investigators conducted 3 FGDs, one with NRC staff another with programme staff of district TB control programme who works either at public health institution (PHI) or at Designated microscopic centers (DMC) and one FGD with STS, whose main work is to register the TB patient and make home visits to those patients to ensure the drug adherence. Investigators invited the participants of one rank in same FGD so that the participants feel free to discuss the topic among them. At the same time to ensure maximum heterogeneity in the response we invited at least one participant from every PHI/DMC. 

For the KII also we selected one staff nurse, one District TB Officer and one NRC in charge Pediatrician to ensure the maximum heterogeneity in the responses and also to represent all the important stakeholders involve in the screening, diagnosis and treatment of childhood TB among children admitted at the NRCs. (Page 7 line 190-199 of revised manuscript with track changes)

COMMENT

Data analysis section: the main concern is that your paper’s title focuses on “TB Screening” (which you explained the five steps of doing that in the methods), but then your main output is only microbiologic confirmation. These are two different things….screening vs confirmation of disease. Thus, your main output is not actually a screening output (if screening was done or not in the children). Rather, your main output is microbiological confirmation of TB disease in these children. Very different things. And as you mention in your Introduction, confirmation of TB in young children (especially when SAM) is very difficult and rare. 

RESPONSE

Thanks for your comment which has provided us invaluable insight regarding the study. The main concern of our paper is poor adherence to TB diagnostic guidelines. For the sake of completion, we have also mentioned how many were microbiologically confirmed among those tested, among total diagnosed (clinical and microbiologically), how were started on treatment and what was the outcome. Microbiological confirmation among those tested cannot be representative of the true picture of its burden because a large number did not undergo tests for microbiological confirmation. We hope this in fine. 

As the TB control Programme of India advocates every child admitted at the NRC has to be screened through CBNAAT by obtaining gastric aspirate. As these are the SAM children and TB might be a cause or consequence of SAM. 

COMMENT

Results

Qualitative phase section: please include results of all of the screening techniques you mention (i.e. of those screened for XYZ, how many screened positive). Of the Mantoux test, physical exam, suggestive symptoms, CXR, history of contact….we need to know how many screened positive for each of those. 

Also, please then list how many of the positive screens (and how many of the children with SAM) were diagnosed with TB, and how many started treatment.

RESPONSE

Thanks for your suggestion to improve the manuscript. The proportion of SAM children underwent different tests for the screening of TB and proportion of children tested positive with respective tests are described in the result section. (line 253-256 of revised manuscript with track changes)

The number of positive screens & number of children diagnosed as TB is been described in result section. (line 257-60)

COMMENT

Line 225: unclear what “infective training” refers to. Need more specifics. Lack of confidence on TB symptoms, reading CXR, score charts, Mantoux testing, sputa extraction? Need more details on the gap.

RESPONSE

Thanks for your suggestion to improve the manuscript. Sorry for the typo error. We have removed the phrase “ineffective training” and clarified it by replacing it with “Lack of training”. (lines 304 to 307 of revised manuscript with track changes)

COMMENT

Line 230: is there also reluctance to sputum induction? Or is one method preferred over the other (which is very important information to share!!). which method is accepted more?

RESPONSE

Thanks for your suggestion to improve the manuscript. The practice of sputum induction is not very common in the studied NRCs. Ideally, sputum induction & gastic aspirate has almost equal yield but due to some reason the practice of sputum induction was not very common in any of the studied NRC. We have discussed 

COMMENT

Discussion

Lines 282-286. For these five recommendations, can you offer any background or evidence base (citations) that these are effective strategies for your setting? They all seem reasonable, but would strengthen the paper if you can show such interventions have been effective at other settings in India, or globally. 

RESPONSE

Thanks for your suggestion to improve the manuscript. Please refer to lines 394-405 of revised manuscript with track changes (discussion section) where we have addressed this. We have dded additional references from 47 to 53 to justify this.

COMMENT

Lines 294-297: you can remove these from discussion. These were already covered in Methods and Results. Focus instead on main messages of the study (main take away points)

RESPONSE

Thanks for suggestion. We have removed that line and focused on the main take away points. We have specially ensured that we do not repeat the result values in the discussion section.

COMMENT

Line 300-301: you mention “availability of diagnostic equipment” here…but in results, this was not mentioned and not listed as a challenge from the interviews. Therefore, we can not mention it here unless it is referenced above in results. 

RESPONSE

 Thanks for suggestion. Availability of diagnostic equipment was one of our recommendations. This was because the overall the extent of testing of gastric aspirate, Mantoux test and chest radiographs was low. Qualitative data showed that poor leadership of in-charge pediatrician/medical officer was one of the reasons for the absence of facility for chest radiograph and CBNAAT (see lines 315-323 of revised manuscript with track changes). We hope this is fine now.

COMMENT

Figures/Tables

Figures 2 and 3 are excellent, and show the entire TB treatment cascade. I would focus more on highlighting these tables in the text, and focusing on the “leakage” and issues at each step!

RESPONSE

 Thanks for appreciation. We tried to describe the highlights of theses tables in result section with leakage & issues at each step. (line 248 to 264 of revised manuscript with track changes)

COMMENT

Again, your title/paper focuses mostly on TB screening…but you have excellent data along the screening, diagnosis, treatment steps. I would highly recommend you expand the scope of title and paper to include these as well (and not limit it to just screening)

RESPONSE

Thanks for appreciation. We have clarified this point before in response to your earlier comment regarding why we have restricted the title to screening. We have revised it mentioning the key finding (as suggested in another comment of yours)

COMMENT

Table 1

Clarify that “More than 1” actually is only “1-5 years old”

Response: Thanks for suggestion. We have changed the same in the table (Table 1)

COMMENT

Residence: remove this as 84-99% didn’t have it recorded (thus not useful)

RESPONSE

Thanks for the suggestion. This indicates, poor record keeping and is important to highlight. Residence details are important for tracking patients for treatment related purpose as well. Hence, with your permission, we would like to retain it. 

COMMENT

Same for SES of family (if not recorded, not useful to include)

RESPONSE

 Thanks for the suggestion. However, investigators have consensus that this is an important factor which determines the access of NRC services by the community. Despite of this fact, the NRC staff doesn’t even capture it, indicates, poor record keeping. Hence, with your permission, investigators want to retain this.

COMMENT

In Table 1, can you also include TB screening (the 5 items) to show how many of the children had each of those 5 screens, and then how many had positive screens of those 5

RESPONSE

 Thanks for the suggestion, we have added this to the table 1 (Table 1)

COMMENT

Table 2: please also include the % in ( ) after each number. The first three columns are missing the (%) and only list N

RESPONSE

 Thanks for the suggestion, we have added % to the screened and diagnosed column. As admitted column is the denominator, the percentage is not applicable here. (Table 2)

COMMENT

Both tables: put your abbreviations footnote below the table (separate) and not as a final row within the table.

RESPONSE

Thanks for the suggestion, we have changed the table 1 & 2 accordingly (Table 1&2)

REVIEWER 2

(Line numbers in our response are with reference to the revised manuscript with track changes)

COMMENT

Point 1: In the section of Introduction (Page 10, Line 2), the “6.%” was incomplete information. The author should check it and ensure the accuracy of the information.

RESPONSE

Thank you for the comment. We have clarified it in the revised manuscript. It now reads as follwows.

“Globally in 2017, an estimated 10% of the total diagnosed tuberculosis (TB) patients were children (<15 years) but in India this was six percent among reported TB [1].” 

COMMENT

Point 2: In the section of Data collection and sources of data (Page 13, Line 16), the full name of “DMCs” should be given when it first appeared in the text.

RESPONSE

Thank you for the comment. We have provided the full name of DMC. We have gone through the document again and ensured that all abbreviations are mentioned in full on first use.

COMMENT

Point 3: In the section of Result, the format of the number of cases and percentages in Figure 2 should be unified. Besides, some footnotes were inconsistent with the contents in the Figure2 and Figure3.

RESPONSE

Thank you for the comment. We have reviewed as advised and made the corrections

COMMENT

Point 4: Detailed descriptions of tables 1 and 2 cannot be found in the Result section, please check it carefully.

RESPONSE

Thank you for the comment. We have provided detailed descriptions of table 1 and 2 in the results section. Please see lines 248-260 of revised manuscript with track changes.

COMMENT

Point 5: The formats of the statistical tables were not standard, and the contents of footnotes were inconsistent with the contents of the tables.

RESPONSE

Thank you for the comment. In the revised manuscript, We have carefully reviewed the formatting of tables and ensured that they meet the accepted standards. We have carefully reviewed the footnotes of the tables as well.

COMMENT

Point 6: The corresponding percentages were missing after number in Table 2.

RESPONSE

Thank you for the comment. Thanks for the suggestion, we have added % to the screened and diagnosed column. As admitted column is the denominator, the percentage is not applicable here. (Table 2)

COMMENT

Point 7: There were too much description of results in the discussion section. The author should refer to other studies for further discussion.

RESPONSE

Thank you for the comment. In the revised manuscript, we have made sure that we do not repeat the results figures in the narrative text of discussion. We also expanded the discussion by discussing more about – one-third coverage of chest radiograph, why induced sputum was not being used as an alternative to gastric lavage in these settings, looking at other literature (ref 47 to 53) where our recommendations have been found to be useful. Please refer to 349-405 lines of revised manuscript with track changes.

COMMENT

Point 8: There were many spelling errors in the manuscript. The author should check and correct it.

RESPONSE

Thank you for the comment. Two authors have carefully gone through the manuscript again and corrected all the typo and grammar errors. Sorry that this happened in the first place.

---

## [Decision Letter · Decision Letter 1]

14 Dec 2020

PONE-D-20-30406R1

Poor adherence to TB diagnosis guidelines among under-five children with severe acute malnutrition in central India: a missed window of opportunity?

PLOS ONE

Dear Dr. Dhingra,

Thank you for submitting your manuscript to PLOS ONE. After careful consideration, we feel that it has merit but does not fully meet PLOS ONE’s publication criteria as it currently stands. Therefore, we invite you to submit a revised version of the manuscript that addresses the points raised during the review process.

Please submit your revised manuscript. If you will need more time than this to complete your revisions, please reply to this message or contact the journal office at plosone@plos.org. Please include the following items when submitting your revised manuscript:

We look forward to receiving your revised manuscript.

Kind regards,

Frederick Quinn

Academic Editor

PLOS ONE

Reviewers' comments:

Reviewer's Responses to Questions

**Comments to the Author**

1. If the authors have adequately addressed your comments raised in a previous round of review and you feel that this manuscript is now acceptable for publication, you may indicate that here to bypass the “Comments to the Author” section, enter your conflict of interest statement in the “Confidential to Editor” section, and submit your "Accept" recommendation.

Reviewer #1: (No Response)

Reviewer #2: All comments have been addressed

2. Is the manuscript technically sound, and do the data support the conclusions?

Reviewer #1: Yes

Reviewer #2: Yes

3. Has the statistical analysis been performed appropriately and rigorously? 

Reviewer #1: Yes

Reviewer #2: Yes

4. Have the authors made all data underlying the findings in their manuscript fully available?

Reviewer #1: Yes

Reviewer #2: Yes

5. Is the manuscript presented in an intelligible fashion and written in standard English?

Reviewer #1: Yes

Reviewer #2: Yes

6. Review Comments to the Author

Reviewer #1: • Overall: The authors have done an excellent job addressing prior review comments and concerns, and the updated manuscript is much improved with a more comprehensive and cohesive message. There are still multiple minor concerns (listed below) that need addressed, but overall the manuscript is vastly improved.

• Cover Page - Abstract:

o Please fix various typographical and grammar errors in the abstract (e.g. extra spaces around “( SAM)”), etc, unnecessary quotation marks, etc)

o In Results, of the 87 will documented outcomes, please also list of those 87 how many had good outcomes (completed treatment, cured, etc) and how many had poor outcomes (died, LTFU, relapse, etc).

• Abstract

o As above, please include the favorable/unfavorable outcomes of the 87 with recorded outcomes

• Introduction:

o Line 71, when mentioning “As per guidelines,…” please specify which guidelines (India national, WHO, India NTP, etc)

• Methods

o Page 4, line 103: please remove the extra space around “ ; “ typo

o Page 5, line 130: Please describe why only Sagar district had qualitative phase (why not also qualitative phase in Sheopur as well?).

o Page 6, line 137-138: it is mentioned a mobile-based questionnaire was used in Feb-May 2018….however, this paragraph is describing the quantitative phase. Was a questionnaire and mobile app used during the quantitative phase? Or for the qualitative phase?? Also, the quantitative phase extracted data from 2017…but the mobile-based questionnaire took place in 2018…need to clarify this date discrepancy.

o Likewise, for the qualitative (interviews) phase…please list the dates these were conducted in the “Data collection and sources of data” section.

• Results

o Page 8, line 188: please again list the dates of the study (to help remind your readers) at the start of the first sentence, e.g. “Between 1st Jan and 31st December 2017, a total of 3230 SAM…”

o Page 8, paragraph of lines 192-196. Need to use alternate terms other than “positive” when discussing symptoms suggestive of TB and chest xray results. It is not commonly terminology to say “positive” for these two screening techniques. Instead, you can say “202 were found to have symptoms suggestive of TB” and “159 had chest xray results suggestive of pTB” respectively.

o Page 8, line 195: please recheck the calculations for the CXR %. 159 is not 1% of 1220 for CXR. Likewise, it is unclear what 485 is 2% of (what is the denominator here?)

o Page 8, lines 197-200: interestingly, as written, your yield of CBNAAT (24%) is much lower than your yield of smear (44%). This is the opposite of the commonly accepted data that Xpert has HIGHER yield than smear in children with TB. If correct, you need to discuss why smear performed so much better than CBNAAT in this cohort (which is highly unusual).

Also, please re-word what is meant by “28 children were labeled as clinically confirmed TB.” By WHO definition, “confirmed” means that a sample was found to have MTb (e.g. sputum). Clinically-diagnosis of TB (without “confirmed”) means that a diagnosis of TB was made based on S/Sx, physical exam, CXR, TST, etc…but without confirmation by Xpert/Culture/smear. If you remove “confirmed” here, it will be ok.

o Page 8 line 201, please add a descriptor to the “Of 3230,” (e.g. “Of the 3230 diagnosed with TB,…”)

o Page 8, line 205-206. Is any information available on the 6% of child with SAM and TB who were not started on ATT? Died? LTFU? Why not started? Also, for the 87 with outcome data, please also include if it was favorable or unfavorable outcome.

o Page 8, line 206-208: the total in this line equals 209, but not the 223 of children with TB.

o Page 10: Figure 2 is not included (only the title and the footnotes are currently in the clean revisions manuscript). Please include the actual Figure 2.

• Discussion

o Page 12, line 284, please remove the comma after “Even though,” (not needed)

o Page 12, line 286, please add “According to our findings,” before the “Adherence to initial assessment…” sentence (as this is what your findings showed)

o Page 13, line 330-331, would suggest removing the “particularly affective and psychomotor domain” portion of sentence, as neither of those was studied or reported in your current study. Better to keep your conclusions focused and linked to your study findings (and not introduce too many new ideas suddenly in the discussion).

o Line 332: need a period at the end of the sentence (after references 40-44).

o Line 333-334: would rephrase that “when samples were collected, a high proportion were collected via gastric aspirate” since overall, a low percentage of children with SAM had any sample collected; but of the few with samples it was done via gastric aspiration (i.e. preferred method).

o Line 335-337: would remove this sentence about transport of specimens, as again, this was not a focus of this study, nor mentioned in Results section, nor was MDR part of this study.

o Line 347, change to “this present study documented poor adherence to initial TB screening, assessment, and TB diagnostic guidelines…”

o General suggestion: throughout the discussion, the theme of putting blame on the NRC in-charge (i.e. a top-down approach) is prevalent throughout. While it may have been reported during the qualitative interviews, I would caution you and the authors on your persistent focus of putting such blame on the in-charge….since both TB and SAM care for children requires a team approach and teamwork mentality to provide the highest quality of comprehensive care. Each clinician and NRC team member should be empowered with responsibility and motivated to have ownership of their actions and of the care they provide. I worry a bit that the recurrent “blame the in-charge” message could be inappropriately used as a way to absolve other team members of blame and of suboptimal care. I would highly suggest you consider alternative way to re-phrase your findings so that instead of singular blame, the message is to better inspire team work and foster personal responsibility to provide the best care possible (even in challenging work situations with limited resources)…rather than simply blaming and pointing the finger at the in-charge.

• Figures/Tables

o Figure 1 and Figure 2: the image quality is poor/blurry. Please use a higher definition image (higher quality) that is legible. Also, there is a blue rectangle pixilation in the upper right (typo? Screen shot artifact?) that needs removed.

There is also a superscript “1” in the “Identified by community & others” box that has no definition at the bottom of the figure to show what the “1” refers to

For the % used, please be consistent with your decimal places. Some % have no decimal point (no tenths of a percentage), while others do use a decimal point (and a tenth of percentage). Also, several boxes do not have any (%) listed (e.g. “referred/died”. Please have consistency throughout the figure.

Reviewer #2: Comments to the Author:

Point 1: In the method section (Page 5, Line 111), the author did not explain how to use the 5 criteria to diagnose tuberculosis. Please check it.

Point 2: In the result section, the description of table1 and figure1 was illogical, the author should rearrange the structure of description.

Point 3: In the result section (Page 8, Line 192), the description “of them 178 (7%) tested positive” was inconsistent with description of Mantoux test in table 1. The author should correct it.

Point 4: The total column of “Time-gap between NRC admission and DOT initiation” in the Table 1 was not necessary. Please check it.

Point 5: In the result section (Page 12, Line 278), the full name of “FNAC” should be given when it first appeared in the text. Please check it carefully.

Point 6: In the footnotes of figures, we could not find “AFP” and “SSM” in the figure 1 and figure 2. The author should check it carefully.

7. PLOS authors have the option to publish the peer review history of their article (what does this mean?). If published, this will include your full peer review and any attached files.

Reviewer #1: No

Reviewer #2: No

---

## [Author Response · Author response to Decision Letter 1]

22 Dec 2020

RESPONSE TO COMMENTS

Review Comments to the Author

Reviewer #1: 

Comment 1: Overall: The authors have done an excellent job addressing prior review comments and concerns, and the updated manuscript is much improved with a more comprehensive and cohesive message. There are still multiple minor concerns (listed below) that need addressed, but overall the manuscript is vastly improved.

Response: Thank you for the constructive comments. We have carefully responded and addressed your second round of comments. 

Comment 2: Cover Page - Abstract:

o Please fix various typographical and grammar errors in the abstract (e.g. extra spaces around “( SAM)”), etc, unnecessary quotation marks, etc)

Response: Thank you for appreciation. Two authors have gone through the manuscript again and corrected the grammar and type errors. 

Comment 3: In Results, of the 87 will documented outcomes, please also list of those 87 how many had good outcomes (completed treatment, cured, etc) and how many had poor outcomes (died, LTFU, relapse, etc).

Response: Thank you for suggestion. As the qualitative phase did not explore the reasons for these poor outcomes, we have decided to restrict the results to how many completed treatment and how many did not have a record of their treatment outcome. We hope this is fine. We have done this so that the focus of the abstract remains on TB screening and barriers around it. The line in the revised manuscript now reads as follows 

“The treatment outcome was favourable (cure or treatment completed) for 70(31%) and not recorded for 121(54%).” (Page 2 line 49-50 of revised manuscript with track changes)

Comment 4: Abstract

o As above, please include the favorable/unfavorable outcomes of the 87 with recorded outcomes

Response: Thank you for suggestion. 

As the qualitative phase did not explore the reasons for these poor outcomes, we have decided to restrict the results to how many completed treatment and how many did not have a record of their treatment outcome. We hope this is fine. We have done this so that the focus of the abstract remains on TB screening and barriers around it. The line in the revised manuscript now reads as follows 

“The treatment outcome was favourable (cure or treatment completed) for 70(31%) and not recorded for 121(54%).” (Page 2 line 49-50 of revised manuscript with track changes)

Comment 5: Introduction:

o Line 71, when mentioning “As per guidelines,…” please specify which guidelines (India national, WHO, India NTP, etc)

Response: Thank you for suggestion. We made the changes (Page 3 line 148 of revised manuscript with track changes)

Comment 6: Methods

o Page 4, line 103: please remove the extra space around “ ; “ typo

Response: Thank you for suggestion. We made the changes (Page 4 line 186 of revised manuscript with track changes)

Comment 7: Page 5, line 130: Please describe why only Sagar district had qualitative phase (why not also qualitative phase in Sheopur as well?).

Response: Thank you for your query. The reason was Concerned staff members were not available in Sheopur district because of NHM workers strike during the qualitative study period (June 2018) and limited funds were available under the project, hence only one district was selected. (Page 5 line 214-216 of revised manuscript with track changes)

Comment 8: Page 6, line 137-138: it is mentioned a mobile-based questionnaire was used in Feb-May 2018….however, this paragraph is describing the quantitative phase. Was a questionnaire and mobile app used during the quantitative phase? Or for the qualitative phase?? Also, the quantitative phase extracted data from 2017…but the mobile-based questionnaire took place in 2018…need to clarify this date discrepancy.

Response: A mobile-based questionnaire was used to retrieve the quantitate phase of data. The data pertaining to Jan-Dec 2017 was retrospectively retrieved from Feb – May 2018. After analysis of quantitative data, qualitative phase of study was performed in June 2018. 

We have clarified in study design that the quantitative phase was a descriptive study involving secondary data collection. Under study population we have clarified that we included under-five children with SAM that were admitted in 2017. Under data collection, we have clarified that secondary data was extracted from the records during Feb-May 2018. Qualitative data collection was done in June 2018. We hope this is fine. 

Please see Page 6 lines 235-237, 241 in revised manuscript with track changes.

Comment 9: Likewise, for the qualitative (interviews) phase…please list the dates these were conducted in the “Data collection and sources of data” section.

Response: Thank you. The qualitative phase of data collection was done in June 2018. (Page 6 line 241 of revised manuscript with track changes)

Comment 10: Results

o Page 8, line 188: please again list the dates of the study (to help remind your readers) at the start of the first sentence, e.g. “Between 1st Jan and 31st December 2017, a total of 3230 SAM…”

Response: Thank you for suggestion. We would not like to repeat the dates again in the results sections as this will be a repletion and may cause reader fatigue. These details have been clarified under study population in methods section. We hope this is fine. 

Comment 11: Page 8, paragraph of lines 192-196. Need to use alternate terms other than “positive” when discussing symptoms suggestive of TB and chest xray results. It is not commonly terminology to say “positive” for these two screening techniques. Instead, you can say “202 were found to have symptoms suggestive of TB” and “159 had chest xray results suggestive of pTB” respectively.

Response: Thank you for suggestion. We made the change (Page 8 line 374-379 of revised manuscript with track changes)

Comment 12: Page 8, line 195: please recheck the calculations for the CXR %. 159 is not 1% of 1220 for CXR. Likewise, it is unclear what 485 is 2% of (what is the denominator here?)

Response: Thank you for suggestion. We have made the changes and clarified the same. We made the change (Page 8 line 377-378 of revised manuscript with track changes)

Comment 13: Page 8, lines 197-200: interestingly, as written, your yield of CBNAAT (24%) is much lower than your yield of smear (44%). This is the opposite of the commonly accepted data that Xpert has HIGHER yield than smear in children with TB. If correct, you need to discuss why smear performed so much better than CBNAAT in this cohort (which is highly unusual).

Response: Thank you for suggestion. In the context of poor record keeping and poor coverage of TB screening, these two percentages are not comparable. We tried to explain this in the first para of discussion section (Page 12 line 757-759 of revised manuscript with track changes)

Comment 14: Also, please re-word what is meant by “28 children were labeled as clinically confirmed TB.” By WHO definition, “confirmed” means that a sample was found to have MTb (e.g. sputum). Clinically-diagnosis of TB (without “confirmed”) means that a diagnosis of TB was made based on S/Sx, physical exam, CXR, TST, etc…but without confirmation by Xpert/Culture/smear. If you remove “confirmed” here, it will be ok.

Response: Thank you for suggestion. We made the change. (Page 8 line 383-384 of revised manuscript with track changes)

Comment 15: Page 8 line 201, please add a descriptor to the “Of 3230,” (e.g. “Of the 3230 diagnosed with TB,…”)

Response: Thank you for suggestion. We made the change. (Page 8 line 385 of revised manuscript with track changes)

Comment 16: Page 8, line 205-206. Is any information available on the 6% of child with SAM and TB who were not started on ATT? Died? LTFU? Why not started? Also, for the 87 with outcome data, please also include if it was favorable or unfavorable outcome.

Response: Thank you for suggestion. We couldn’t explore the reason for the same. As there was no record available at DMC near to NRC. It seems poor coordination between NRC & RNTCP staff. Ideally, these children should have been notified during their stay at NRC itself and same should have been mentioned in their Case sheet of NRC. We included the outcome for those it was documented. (Page 8 line 390-391 of revised manuscript with track changes)

Comment 17: Page 8, line 206-208: the total in this line equals 209, but not the 223 of children with TB.

Response: Thank you for suggestion. We made the change. (Page 8 line 389-391 of revised manuscript with track changes)

Comment 18: Page 10: Figure 2 is not included (only the title and the footnotes are currently in the clean revisions manuscript). Please include the actual Figure 2.

Response: As per PLOS ONE guidelines, after we cite a figure in result narrative, we are required to insert the figure title and footnote. However the Figure itself has to be submitted separately in the form of an image file (tiff format). In other words, the figure is not part of the manuscript file. However, the reviewer can access the figure file in the submission pdf. 

Comment 19: Discussion

o Page 12, line 284, please remove the comma after “Even though,” (not needed)

Response: Thank you for suggestion. We made the change. (Page 12 line 756 of revised manuscript with track changes)

Comment 20: Page 12, line 286, please add “According to our findings,” before the “Adherence to initial assessment…” sentence (as this is what your findings showed)

Response: Thank you for suggestion. We made the change. (Page 12 line 760 of revised manuscript with track changes)

Comment 21: Page 13, line 330-331, would suggest removing the “particularly affective and psychomotor domain” portion of sentence, as neither of those was studied or reported in your current study. Better to keep your conclusions focused and linked to your study findings (and not introduce too many new ideas suddenly in the discussion).

Response: Thank you for suggestion. We made the change. (Page 14 line 841 of revised manuscript with track changes)

Comment 22: Line 332: need a period at the end of the sentence (after references 40-44).

Response: Thank you for suggestion. We made the change. (Page 14 line 843 of revised manuscript with track changes)

Comment 23: Line 333-334: would rephrase that “when samples were collected, a high proportion were collected via gastric aspirate” since overall, a low percentage of children with SAM had any sample collected; but of the few with samples it was done via gastric aspiration (i.e. preferred method).

Response: Thank you for suggestion. We have actually revised the sentence as follows “Low coverage of collection and testing of gastric aspirate might be due to the lack of assured specimen transport mechanism to the CBNAAT testing facilities at the district headquarters” (Page 14 line 843-844 of revised manuscript with track changes)

Comment 24: Line 335-337: would remove this sentence about transport of specimens, as again, this was not a focus of this study, nor mentioned in Results section, nor was MDR part of this study.

Response: Thank you for suggestion. We made the change in this sentence as per above comments, didn’t removed the sentence, we hope reviewer will find it OK.. (Page 14 line 843-844 of revised manuscript with track changes)

Comment 25: Line 347, change to “this present study documented poor adherence to initial TB screening, assessment, and TB diagnostic guidelines…”

Response: Thank you for suggestion. We made the change. (Page 14 line 854 of revised manuscript with track changes)

Comment 26: General suggestion: throughout the discussion, the theme of putting blame on the NRC in-charge (i.e. a top-down approach) is prevalent throughout. While it may have been reported during the qualitative interviews, I would caution you and the authors on your persistent focus of putting such blame on the in-charge….since both TB and SAM care for children requires a team approach and teamwork mentality to provide the highest quality of comprehensive care. Each clinician and NRC team member should be empowered with responsibility and motivated to have ownership of their actions and of the care they provide. I worry a bit that the recurrent “blame the in-charge” message could be inappropriately used as a way to absolve other team members of blame and of suboptimal care. I would highly suggest you consider alternative way to re-phrase your findings so that instead of singular blame, the message is to better inspire team work and foster personal responsibility to provide the best care possible (even in challenging work situations with limited resources)…rather than simply blaming and pointing the finger at the in-charge.

Response: Thank you for suggestion. We change the phrase to “poor team skills” in fig 2. 

Comment 27: Figures/Tables

o Figure 1 and Figure 2: the image quality is poor/blurry. Please use a higher definition image (higher quality) that is legible. Also, there is a blue rectangle pixilation in the upper right (typo? Screen shot artifact?) that needs removed.

Response: Thank you for suggestion. We made the change. (Fig 1 & 2)

Comment 28: There is also a superscript “1” in the “Identified by community & others” box that has no definition at the bottom of the figure to show what the “1” refers to

Response: Thank you for suggestion. We remove this. (Fig 1 & 2)

Comment 29: For the % used, please be consistent with your decimal places. Some % have no decimal point (no tenths of a percentage), while others do use a decimal point (and a tenth of percentage). Also, several boxes do not have any (%) listed (e.g. “referred/died”. Please have consistency throughout the figure.

Response: Thank you for suggestion. We made the change. (Fig 1 & 2)

Reviewer #2: 

Comments to the Author:

Point 1: In the method section (Page 5, Line 111), the author did not explain how to use the 5 criteria to diagnose tuberculosis. Please check it.

Response: Thank you for suggestion. If microbiological diagnosis is not feasible, the in-charge pediatrician or Medical Officer can make a clinical diagnosis depending on the initial assessment using 5 criteria and further investigations of examination findings. (Page 5 line 195-208 of revised manuscript with track changes)

Point 2: In the result section, the description of table1 and figure1 was illogical, the author should rearrange the structure of description.

Response: Thank you for suggestion. We made the changes. (Table 1 & Fig 1)

Point 3: In the result section (Page 8, Line 192), the description “of them 178 (7%) tested positive” was inconsistent with description of Mantoux test in table 1. The author should correct it.

Response: Thank you for pointing our mistake. We made the correction. (Page 8 line 374 of revised manuscript with track changes)

Point 4: The total column of “Time-gap between NRC admission and DOT initiation” in the Table 1 was not necessary. Please check it.

Response: Dear Reviewer, we have deleted it from table 1

Point 5: In the result section (Page 12, Line 278), the full name of “FNAC” should be given when it first appeared in the text. Please check it carefully.

Response: Thank you for suggestion. Actually, we have removed FNAC as our quote does not discuss about it. (Page 12 line 750 of revised manuscript with track changes)

Point 6: In the footnotes of figures, we could not find “AFP” and “SSM” in the figure 1 and figure 2. The author should check it carefully.

Response: Thank you for suggestion. We made the changes

---

## [Decision Letter · Decision Letter 2]

4 Jan 2021

PONE-D-20-30406R2

Poor adherence to TB diagnosis guidelines among under-five children with severe acute malnutrition in central India: a missed window of opportunity?

PLOS ONE

Dear Dr. Dhingra,

Thank you for submitting your manuscript to PLOS ONE. After careful consideration, we feel that it has merit but does not fully meet PLOS ONE’s publication criteria as it currently stands. Therefore, we invite you to submit a revised version of the manuscript that addresses the points raised during the review process.

Please submit your revised manuscript. If you will need significantly more time to complete your revisions, please reply to this message or contact the journal office at plosone@plos.org. Please include the following items when submitting your revised manuscript:

We look forward to receiving your revised manuscript.

Kind regards,

Frederick Quinn

Academic Editor

PLOS ONE

Reviewers' comments:

Reviewer's Responses to Questions

**Comments to the Author**

1. If the authors have adequately addressed your comments raised in a previous round of review and you feel that this manuscript is now acceptable for publication, you may indicate that here to bypass the “Comments to the Author” section, enter your conflict of interest statement in the “Confidential to Editor” section, and submit your "Accept" recommendation.

Reviewer #1: (No Response)

Reviewer #2: All comments have been addressed

2. Is the manuscript technically sound, and do the data support the conclusions?

Reviewer #1: Partly

Reviewer #2: Yes

3. Has the statistical analysis been performed appropriately and rigorously? 

Reviewer #1: Yes

Reviewer #2: Yes

4. Have the authors made all data underlying the findings in their manuscript fully available?

Reviewer #1: Yes

Reviewer #2: Yes

5. Is the manuscript presented in an intelligible fashion and written in standard English?

Reviewer #1: Yes

Reviewer #2: Yes

6. Review Comments to the Author

Reviewer #1: • Overall: The authors have done a good job addressing certain comments. However, there are still issues and gaps that need addressed to make the manuscript stronger and flow more logically and coherently. There are also some data discrepancies between results and discussion that need addressed.

• Abstract

o In Background, you specifically mention “presumptive pediatric pulmonary TB patients” however the title and rest of the abstract simply uses “TB” (without the ‘pulmonary’ preface). Throughout your study, are you only reviewing/reporting on pulmonary TB? Or does your data include all types of child TB (pTB and EPTB)? Whatever the case, either remove “pulmonary” here, or add “pulmonary” throughout so it is clear and harmonized in the paper (“pTB” vs “TB”).

o In Objective, remove “(2017)” and instead change to “in 2017.”

o Methods, remove the “Between February and May 2018”, as it is not critical to report when the analysis was done. Rather, it is much more important to know the dates of retrospective review (Jan-Dec 2017). Writing when the study teamed reviewed the 2017 data is unnecessary (and could add to some date confusion of the reader). Instead, you can start the Methods with “An explanatory mixed methods study was conducted to analyze the 2017 child TB data among under-fives with SAM at NRCs in Sagar and Sheopur districts. The NRC records were reviewed,….”

o Methods, your definition of “adherence to TB diagnosis guidelines” as listed in the abstract is very narrow (i.e. it only focuses on specimen collection). TB diagnostic guidelines also involve clinical history suggestive of TB, physical exam findings suggestive of TB, lack of response to other non-TB treatments, and other non-sputum/molecular tests (e.g. radiography/CXR, IGRA, etc). When assessing “adherence to TB diagnostic guidelines” (yes/no), did you team evaluate all of these diagnostic pathways? Or just whether a specimen for AFB smear and/or Xpert was obtained? If the former, please clarify this in the abstract.

o Results, in your abstract, you provide no results for the “Operationally, adherence to TB diagnosis guidelines” (yes/no) that was mentioned in your Methods. Please lists the data on what %/N were “yes” for adhered to guidelines. Your results only mention TB screening and gastric aspirates….which are a part/subset of “TB diagnostic guidelines” but are not comprehensive to TB diagnosis.

o In Results, “poor team skills” as written is unclear and not a commonly used term. I know this is in response to an earlier comment, but needs rephased/rewritten so that it is easily understandable to your audience what you are trying to convey as a challenge. Telling someone “poor team skills” is a challenge is hard to interpret what exactly you mean.

o Conclusion: needs rephrased. The single sentence as written is difficult to understand. For example, you write “its utilization remains suboptimal” yet the subject of the sentence is NRCs….thus, is NRC utilization suboptimal? Or do you mean that following TB screening and diagnostic guidelines within the NRCs are suboptimal? This conclusion in the abstract needs rewritten to better summarize the findings of your study.

• Introduction:

o Line 63: here you mention the high rates of EPTB in child TB; to my earlier comment above…is your study only looking at pTB or all types of TB? If only pTB, it seems odd to mention the higher rates of EPTB in children…but then not to analyze EPTB in your cohort. Line 82 specifically mentioned “pediatric pulmonary TB” again. What about children with EPTB? You need to make it abundantly clear if you are only assessing pTB or all types TB in your cohort of under-fives with SAM.

o Line 76 and line 80. Add appropriate spaces after the period (before “In”).

o Lines 83-85. Please add in a sentence or details linking the gaps in India’s NRCs to what you aimed to do with your study. E.g. “In order to try and better address and understand the reason for these large TB treatment gaps in children with SAM, our study aimed to estimate the yield…” Otherwise, the final two sentences do not flow well from the rest of your Introduction.

• Methods

o Line 139, please delete “between Feb and May 2018” as it is not necessary to report when your study team did their data extraction/analysis. What is important is to share the dates the data was extracted from (2017), which you have already done. The 2018 dates could add to date confusion for the audience.

• Results

o Line 196, add a space after “(6%)”

o Line 212-213: there are 209 who started TB treatment, but you only have 70 with favorable and 121 not recorded…which is 191 total. What about the other 18 patients (of the 209)?? Your outcomes (since they include “not recorded” should tally the 209 who started ATT

o Again, I am having trouble on how you are defining “Screened for TB” in your Table 2. It says only 17% screened for TB in this table. But in your text above, it mentions 83% had Mantoux text, 70% were asked about symptoms of TB….and both of these are ways to screen for TB. From your text, it looks like upwards of 70-83% were screened for TB (in one way or another), but your table says only 17%? This is confusing to the reader and needs clarified and addressed. I think your table means to show who had smear or Xpert…which is different from “Screened as per guidelines”, I would recommend removing “screened as per guidelines” and instead write “Underwent smear microscopy or rapid molecular test”

o Line 261 and section: again, consider different terminology. “poor team skills” is not commonly used, and vague and hard to understand as written. Lack of motivation, lack of ownership, or work avoidance seem better terms to use what I think you are trying to describe.

o Line 273: I don’t think “ambiguity” is the correct term for the challenge here. As written, it sounds like the way the Indian TB guidelines are written is not clear and ambiguous. Is that the issue? Guidelines are not clearly written? Or is it again a type of “work avoidance” where team members don’t want to have the responsibility of diagnosis and treating TB in children?

o Line 279: “Non-availability” or “Unavailable” would be more appropriate terms to use here instead of “Non-feasibility”

• Discussion

o Line 285-286: as mentioned above in Abstract, this first sentence needs rewritten for clarification and ease of understanding. It makes it sound like NRC utilization is suboptimal as written.

o Line 287: change “coverage” to “implementation”

Also, I still struggle to understand how your smear yield is so much better and so much higher than your Xpert/CBNAAT yield. The added text doesn’t help explain this. Can you elaborate more on why this mis-matched performance of AFB smear vs Xpert in your cohort?

o Line 288: were diagnostic guidelines different? Or do you mean they way they were implemented (or lack of implementation at sites) is what differed? I think you may want to change “followed” to “implemented”

o Line 300: here you say less than one-third…but in your results (line 198 ) you say 2277/70% were asked about symptoms suggestive of TB. This wide data discrepancy needs to addressed and aligned.

o Line 309-310: again, you say one-third had CXR and Mantoux test…but in your results (line 196), there were 2665/83% with Mantoux testing. Again, your Results and your Discussion need to align and harmonize.

o Line 310-311: recommend changing “availability” to “usage” or “implementation” as the availability or stock out of such test are not related to teamwork or motivation (but usage is)

o In your Discussion, many paragraphs talk about the quantitative findings (and interpreting them). But there really aren’t any paragraphs in your discussion that talk about and interpret the various qualitative (interview) findings of your study that you present in your results. Before jumping into your “The following are our recommendations” paragraph, it would be good to discuss and interpret the findings of your qualitative phases (the interviews) in your Discussion. Doing this would also make “your recommendations” flow more naturally and be derived from the findings in your Results. Likewise, in line 336 you mention how the qualitative phase was “invaluable in complementing”; yet the qualitative findings receive little to no attention in your discussion

o Line 336: add a spacing before this new paragraph.

o

• Figures/Tables

o Figure 1 and Figure 2: the image quality is still very poor/blurry in the built pdf. It is difficult to read due to blurriness, and needs a higher quality image that is legible.

Reviewer #2: Comments to the Author:

Point 1: In the result section, the first paragraph described the contents of table 2, not Table 1. Please check it carefully.

Point 2: The contents of table 2 was more suitable described firstly in the Quantitative phase, the author should set table 2 to table 1.

Point 3: In the Quantitative phase section, the description of tables and figures is illogical. The author should describe them as following figuere1, table2 and table1.

Point 4: We still could not find “AFP” and “SSM” in the footnote of figure 1. Please check it carefully.

Point 5: The image of figure1 and figure2 was blurry. Please use a higher definition image.

7. PLOS authors have the option to publish the peer review history of their article (what does this mean?). If published, this will include your full peer review and any attached files.

Reviewer #1: No

Reviewer #2: No

---

## [Author Response · Author response to Decision Letter 2]

9 Feb 2021

RESPONSE TO REVIEWERS COMMENTS

Reviewer #1:

Comment 

 • Overall: The authors have done a good job addressing certain comments. However, there are still issues and gaps that need addressed to make the manuscript stronger and flow more logically and coherently. There are also some data discrepancies between results and discussion that need addressed.

Response: Thank you for the appreciation. We tried our best to address the issues and gaps suggested by the reviewer in an order to improve the manuscript. Based on your comments we have also removed any data discrepancies between result and discussion section.

Comment 

• Abstract

o In Background, you specifically mention “presumptive pediatric pulmonary TB patients” however the title and rest of the abstract simply uses “TB” (without the ‘pulmonary’ preface). Throughout your study, are you only reviewing/reporting on pulmonary TB? Or does your data include all types of child TB (pTB and EPTB)? Whatever the case, either remove “pulmonary” here, or add “pulmonary” throughout so it is clear and harmonized in the paper (“pTB” vs “TB”).

Response: Our study intended to include all forms of TB. We have removed “pulmonary” wherever applicable. We have ensured consistency in the revised manuscript. 

All children with SAM that are admitted to NRCs were expected to undergo TB screening. This includes assessment for microbiological confirmation (using upfront rapid molecular test, operationally we included sputum microscopy as well) and clinical assessment (history suggestive to TB, history of contact, physical examination, chest x ray & Mantoux test). 

Comment 

o In Objective, remove “(2017)” and instead change to “in 2017.”

Response: Thanks for suggestion. We made the change. (Line 37 & 95 in Revised Manuscript with track changes)

Comment 

o Methods, remove the “Between February and May 2018”, as it is not critical to report when the analysis was done. Rather, it is much more important to know the dates of retrospective review (Jan-Dec 2017). Writing when the study teamed reviewed the 2017 data is unnecessary (and could add to some date confusion of the reader). Instead, you can start the Methods with “An explanatory mixed methods study was conducted to analyze the 2017 child TB data among under-fives with SAM at NRCs in Sagar and Sheopur districts. The NRC records were reviewed,….”

Response: Thanks for suggestion. The cohort of under five children admitted to NRC belonged to 2017 (based on date of admission). Between Feb 2018 and May 2018 the records of these children were reviewed. I think this is important to clarify. Including the time when data extraction was done was also suggested by reviewer 2, we agreed with the suggestions and included this information during revision. We need to give some time of (retrospective) follow up for those admitted in the end of Dec 2017 before we reviewed their data. Hence, we started the review of records in Feb 2018. This is the reason why some children were still under treatment (line 228-29 of revised manuscript with track changes). We have not mentioned Feb-May 2018 in abstract but mentioned it under data collection in main text. We hope this is fine. 

The study design reads as follows (lines 101-02 of revised manuscript with track changes)

“An explanatory type of mixed methods study design was used [27]. The descriptive quantitative phase involving secondary data collection was followed by a descriptive qualitative phase.”

The data collection line reads as follows (line 149-51 of revised manuscript with track changes)

“In the quantitative phase, between February and May 2018, information related to screening, diagnosis and treatment was extracted and captured in a mobile based data collection tool [32].”

Comment 

o Methods, your definition of “adherence to TB diagnosis guidelines” as listed in the abstract is very narrow (i.e. it only focuses on specimen collection). TB diagnostic guidelines also involve clinical history suggestive of TB, physical exam findings suggestive of TB, lack of response to other non-TB treatments, and other non-sputum/molecular tests (e.g. radiography/CXR, IGRA, etc). When assessing “adherence to TB diagnostic guidelines” (yes/no), did you team evaluate all of these diagnostic pathways? Or just whether a specimen for AFB smear and/or Xpert was obtained? If the former, please clarify this in the abstract.

Response: Thank you. In this revised manuscript we have clarified that we determined the extent of clinical assessment and assessment for microbiological confirmation among under-five children with SAM admitted at NRC. The number (proportion) undergoing clinical assessment and microbiological confirmation have been presented separately. We have consistently mentioned this in the abstract and main text as well. Please see lines 35-36, 46-50, 95-96, 179-83, line 212-21 in revised manuscript with track changes. I am copy pasting below the following from lines 179-83 of revised manuscript with track changes.

“Key analytic outputs were the number (proportion) of children that were clinically assessed (each criteria presented separately) and the number (proportion) that underwent smear microscopy or rapid molecular tests to establish microbiological confirmation. It was operationally decided to include sputum microscopy for the assessment of microbiological confirmation”

Comment 

o Results, in your abstract, you provide no results for the “Operationally, adherence to TB diagnosis guidelines” (yes/no) that was mentioned in your Methods. Please lists the data on what %/N were “yes” for adhered to guidelines. Your results only mention TB screening and gastric aspirates….which are a part/subset of “TB diagnostic guidelines” but are not comprehensive to TB diagnosis.

Response: We have removed this statement “Operationally, adherence to TB diagnosis guidelines (yes/no)” in main text. In continuation to our responses above, we have clarified that TB screening included clinical assessment (each criteria presented separately) and assessment for microbiological confirmation (CBNAAT or sputum microscopy). Results also have been presented separately for clinical assessment (each criteria presented separately) and assessment for microbiological confirmation (CBNAAT or sputum microscopy). Please see lines 35-36, 46-50, 95-96, 179-83, line 212-21 in revised manuscript with track changes. We hope this is now clear. 

Comment 

o In Results, “poor team skills” as written is unclear and not a commonly used term. I know this is in response to an earlier comment, but needs rephased/rewritten so that it is easily understandable to your audience what you are trying to convey as a challenge. Telling someone “poor team skills” is a challenge is hard to interpret what exactly you mean.

Response: This edit in the phrase of the theme (previously it was poor leadership by NRC in charge) was made during prior revisions based on the suggestion by reviewer 2. Hence, we are retaining this phrase ‘poor team skills’. We hope this is fine. 

Comment 

o Conclusion: needs rephrased. The single sentence as written is difficult to understand. For example, you write “its utilization remains suboptimal” yet the subject of the sentence is NRCs….thus, is NRC utilization suboptimal? Or do you mean that following TB screening and diagnostic guidelines within the NRCs are suboptimal? This conclusion in the abstract needs rewritten to better summarize the findings of your study.

Response: Thank you for the comment. We have revised as follows (line 60-52, 322-24) of revised manuscript

“Though NRCs provided a unique window of opportunity for the screening and management of TB among under-five children with SAM, the utilization of this opportunity remained suboptimal.”

Comment 

• Introduction:

o Line 63: here you mention the high rates of EPTB in child TB; to my earlier comment above…is your study only looking at pTB or all types of TB? If only pTB, it seems odd to mention the higher rates of EPTB in children…but then not to analyze EPTB in your cohort. Line 82 specifically mentioned “pediatric pulmonary TB” again. What about children with EPTB? You need to make it abundantly clear if you are only assessing pTB or all types TB in your cohort of under-fives with SAM.

Response: Our study intended to include all forms of TB and their distributions in terms of how they are being diagnosed i.e. microbiologically confirm TB or clinically confirm TB. We have ensured that we mention this consistently throughout the manuscript. However, we didn’t capture how many of the children with TB had extrapulmonary TB and hence, have not mentioned them in the results. 

Comment 

o Line 76 and line 80. Add appropriate spaces after the period (before “In”).

Response: Thanks for suggestion. We made the change. 

Comment

o Lines 83-85. Please add in a sentence or details linking the gaps in India’s NRCs to what you aimed to do with your study. E.g. “In order to try and better address and understand the reason for these large TB treatment gaps in children with SAM, our study aimed to estimate the yield…” Otherwise, the final two sentences do not flow well from the rest of your 

Response

We have revised this (see lines 93-98 of revised ma manuscript with track changes, reproduced below)

“In order to understand the TB diagnosis gaps in children with SAM, our study aimed to determine the yield of and adherence to the TB diagnostic guidelines (clinical assessment and assessment for microbiological confirmation) among under-five children with SAM admitted at NRCs of Madhya Pradesh, India in 2017. We also explored the challenges in screening from the health care providers’ perspective.”

Comment 

• Methods

o Line 139, please delete “between Feb and May 2018” as it is not necessary to report when your study team did their data extraction/analysis. What is important is to share the dates the data was extracted from (2017), which you have already done. The 2018 dates could add to date confusion for the audience.

Response: Thank you for the suggestion. The cohort of under five children admitted to NRC belonged to 2017 (based on date of admission). Between Feb 2018 and May 2018 the records of these children were reviewed. I think this is important to clarify. This was also suggested by reviewer 2, we agreed with the suggestions and included this information. We need to give some time of (retrospective) follow up for those admitted in the end of Dec 2017 before we reviewed their data. Hence, we started the review of records in Feb 2018. This is the reason why some children were still under treatment (line 230-31 of revised manuscript with track changes). We have not mentioned Feb-May 2018 in abstract but mentioned it under data collection in main text. We hope this is fine. 

The study design reads as follows (lines 101-02 of revised manuscript with track changes)

“An explanatory type of mixed methods study design was used [27]. The descriptive quantitative phase involving secondary data collection was followed by a descriptive qualitative phase.”

The data collection line reads as follows (line 149-51 of revised manuscript with track changes)

“In the quantitative phase, between February and May 2018, information related to screening, diagnosis and treatment was extracted and captured in a mobile based data collection tool [32].”

Comment 

• Results

o Line 196, add a space after “(6%)”

Response: Thanks for suggestion. We made the change. 

Comment 

o Line 212-213: there are 209 who started TB treatment, but you only have 70 with favorable and 121 not recorded…which is 191 total. What about the other 18 patients (of the 209)?? Your outcomes (since they include “not recorded” should tally the 209 who started ATT

Response: Thanks for suggestion. We described the treatment outcome for all 209 children with SAM who initiated on ATT. (Line 228 of Revised Manuscript with tract changes). 

 “Of 209, the treatment outcome was favourable for 70(34%) (cured or treatment completed). Treatment was still going on for 17(8%), outcome was not recorded for 121(58%) and one child failed the treatment”

209 = 70 + 17 +121 +1 

Comment 

o Again, I am having trouble on how you are defining “Screened for TB” in your Table 2. It says only 17% screened for TB in this table. But in your text above, it mentions 83% had Mantoux text, 70% were asked about symptoms of TB….and both of these are ways to screen for TB. From your text, it looks like upwards of 70-83% were screened for TB (in one way or another), but your table says only 17%? This is confusing to the reader and needs clarified and addressed. I think your table means to show who had smear or Xpert…which is different from “Screened as per guidelines”, I would recommend removing “screened as per guidelines” and instead write “Underwent smear microscopy or rapid molecular test”

Response: Thank you for the suggestion, in Table 2, we have replaced ‘screened for TB’ with ‘assessed for microbiological confirmation’. This is also in line with our previous comment that we have removed “Operationally, adherence to TB diagnosis guidelines (yes/no)” in the main text and presented clinical assessment and assessment for microbiological assessment separately.

Comment 

o Line 261 and section: again, consider different terminology. “poor team skills” is not commonly used, and vague and hard to understand as written. Lack of motivation, lack of ownership, or work avoidance seem better terms to use what I think you are trying to describe.

Response: This edit in the phrase of the theme (previously it was poor leadership by NRC in charge) was made based on the suggestion by reviewer 2 during the previous rounds of reviews. Hence, we are retaining this phrase ‘poor team skills’. We hope this is fine. 

Comment 

o Line 273: I don’t think “ambiguity” is the correct term for the challenge here. As written, it sounds like the way the Indian TB guidelines are written is not clear and ambiguous. Is that the issue? Guidelines are not clearly written? Or is it again a type of “work avoidance” where team members don’t want to have the responsibility of diagnosis and treating TB in children?

Response: This was the perception of the health care providers. CBNAAT was not available in all facilities. The screening and diagnostic guidelines were not explicit regarding what was to be done when certain tests/facilities were either not available or not feasible due to lack of assured specimen transport; therefore the screening guidelines were not consistently followed across the NRCs. 

We discussed this again internally. We believe that “non-feasibility of CBNAAT” (previously included as a separate theme) is covered under the description of theme “ambiguity in diagnostic algorithm”. Hence we have deleted the theme “non-feasibility of CBNAAT” from the narrative text as well as Figure 2.

We have clarified this in lines 304-312 of revised manuscript with track changes.

Comment 

o Line 279: “Non-availability” or “Unavailable” would be more appropriate terms to use here instead of “Non-feasibility”

Response: We discussed this again internally. We believe that “non-feasibility of CBNAAT” (previously included as a separate theme) is covered under the description of theme “ambiguity in diagnostic algorithm”. Hence, we have deleted the theme “non-feasibility of CBNAAT” from narrative text as well as Figure 2. CBNAAT was not available in all facilities. The screening and diagnostic guidelines were not explicit regarding what was to be done when certain tests/facilities were either not available or not feasible due to lack of assured specimen transport; therefore the screening guidelines were not consistently followed across the NRCs. 

Comment 

• Discussion

o Line 285-286: as mentioned above in Abstract, this first sentence needs rewritten for clarification and ease of understanding. It makes it sound like NRC utilization is suboptimal as written.

Response: As shared before, we have modified the statement. Thank you for this suggestion. Lines 320-22 of revised manuscript with track changes.

Comment 

o Line 287: change “coverage” to “implementation”

Response: Thanks for suggestion. We made the change. (Line 324 in Revised Manuscript with tract changes)

Comment 

Also, I still struggle to understand how your smear yield is so much better and so much higher than your Xpert/CBNAAT yield. The added text doesn’t help explain this. Can you elaborate more on why this mis-matched performance of AFB smear vs Xpert in your cohort?

Response: In the context of poor implementation of TB screening (akin to high non response in a survey), the yield in our small non-representative sample cannot be extrapolated to all children admitted at NRC. We clarified this in discussion section (see lines 22-25 of revised manuscript with track changes, reproduced below). 

“In the context of poor implementation of TB screening, the yield of smear microscopy and CBNAAT in the small subgroup of children who underwent testing cannot be extrapolated to all the children admitted at NRC (n=3230). The diagnostic guidelines implemented at different NRCs were different”

Comment 

o Line 288: were diagnostic guidelines different? Or do you mean they way they were implemented (or lack of implementation at sites) is what differed? I think you may want to change “followed” to “implemented”

Response: Thanks for suggestion. We made the change. (Line 327 in Revised Manuscript with tract changes)

Comment 

o Line 300: here you say less than one-third…but in your results (line 198 ) you say 2277/70% were asked about symptoms suggestive of TB. This wide data discrepancy needs to addressed and aligned.

Response: Thanks for pointing this mistake. We made the change. (Line 340-42 in Revised Manuscript with tract changes)

Comment 

o Line 309-310: again, you say one-third had CXR and Mantoux test…but in your results (line 196), there were 2665/83% with Mantoux testing. Again, your Results and your Discussion need to align and harmonize.

Response: Thanks for pointing this mistake. We made the change. (Line 350 in Revised Manuscript with tract changes)

Comment 

o Line 310-311: recommend changing “availability” to “usage” or “implementation” as the availability or stock out of such test are not related to teamwork or motivation (but usage is)

Response: Thanks for pointing this mistake. We made the change. (Line 352 in Revised Manuscript with tract changes)

Comment 

o In your Discussion, many paragraphs talk about the quantitative findings (and interpreting them). But there really aren’t any paragraphs in your discussion that talk about and interpret the various qualitative (interview) findings of your study that you present in your results. Before jumping into your “The following are our recommendations” paragraph, it would be good to discuss and interpret the findings of your qualitative phases (the interviews) in your Discussion. Doing this would also make “your recommendations” flow more naturally and be derived from the findings in your Results. Likewise, in line 336 you mention how the qualitative phase was “invaluable in complementing”; yet the qualitative findings receive little to no attention in your discussion

Response: Thanks for the suggestion. We considered your comments but the authors feel that the discussion section is fine in the present form. We have discussed the quantitative and qualitative findings together. This was an explanatory mixed methods study. The findings of the quan phase were explored indepth (why?) in the qual phase. While interpreting the quan findings in the discussion section, we also side by side discussed the relevant qual findings. Hence, we did not find the need to have a separate qual paragraph in the discussion section. We hope this is fine. 

During this revision, we have added relevant heading in the discussion section, this will hopefully; make it easy for the reader. The headings are: key findings, recommendations, strengths and limitations, conclusion

Comment 

o Line 336: add a spacing before this new paragraph.

Response: Thanks. We added the space. 

Comment 

• Figures/Tables

o Figure 1 and Figure 2: the image quality is still very poor/blurry in the built pdf. It is difficult to read due to blurriness, and needs a higher quality image that is legible.

Response: Thanks for pointing our mistake. The image files (Fig 1 and 2) are submitted separately at the time of submission. We have made sure that the image files are clear and meet PLOS ONE requirement. When these image files are merged into the pdf by the manuscript submission website (while the software generates the submission pdf file), it is not uncommon for the figures to become blurred. However, as this comment has been made for the second time by the reviewer, we have remade the figures using MS powerpoint and submitted the more clear image files as figure files. (previously we used insert canvas option in MS word to make the figure files). We have taken this opportunity to make minor edits in Fig 1 and 2 so that they are consistent with the narrative.

Once accepted, the journal editorial office will use the revised Figure files that have been submitted separately (these are clear and as per PLOS ONE guidelines)

Reviewer #2: 

Comment 

Point 1: In the result section, the first paragraph described the contents of table 2, not Table 1. Please check it carefully.

Response: Thanks for pointing our mistake. We have cross checked this and made corrections

Comment 

Point 2: The contents of table 2 was more suitable described firstly in the Quantitative phase, the author should set table 2 to table 1.

Response: Thanks for suggestion. We discussed your comments and the authors decided to have the following order. To maintain the flow of the results section, we have first described the number (proportion) that were clinically assessed (each criteria separately). Then we mention the number (proportion) that undergo assessment for microbiological confirmation and how many TB patients (microbiologically confirmed and clinically confirmed) were eventually diagnosed. Then we talk about the treatment initiation and outcome. We hope this is fine. The tables and figures have been cited accordingly in the results narrative.

Comment 

Point 3: In the Quantitative phase section, the description of tables and figures is illogical. The author should describe them as following figuere1, table2 and table1.

Response: Thanks for suggestion. We made the change. We have started with citing Figure 1. In line with our comments above, we have first described the table with baseline characteristics and clinical assessment. Then we have described the table depicting assessment of microbiological confirmation and TB diagnosis (microbiologically and clinically confirmed). (Line 247-257 in Revised Manuscript with tract changes)

Comment 

Point 4: We still could not find “AFP” and “SSM” in the footnote of figure 1. Please check it carefully.

Response: Thanks for pointing our mistake. We removed “SSM” and corrected AFP to “AFB”. (Fig 1 footnotes) 

Comment 

Point 5: The image of figure1 and figure2 was blurry. Please use a higher definition image.

Response: Thanks for pointing out. The image files (Fig 1 and 2) are submitted separately at the time of submission. We have made sure that the image files are clear and meet PLOS ONE requirement. When these image files are merged into the pdf (while the software generates the submission pdf file), it is not uncommon for the figures to become blurred. However, as this comment has been made for the second time by the reviewer, we have remade the figures using MS powerpoint and submitted the more clear image files as figure files. (previously we used insert canvas option in MS word to make the figure files). We have taken this opportunity to make minor edits in Fig 1 and 2 so that they are consistent with the narrative.

Once accepted, the journal editorial office will use the revised Figure files that have been submitted separately (these are clear and as per PLOS ONE guidelines)

---

## [Decision Letter · Decision Letter 3]

22 Feb 2021

Poor adherence to TB diagnosis guidelines among under-five children with severe acute malnutrition in central India: a missed window of opportunity?

PONE-D-20-30406R3

Dear Dr. Dhingra,

We’re pleased to inform you that your manuscript has been judged scientifically suitable for publication and will be formally accepted for publication once it meets all outstanding technical requirements.

Kind regards,

Frederick Quinn

Academic Editor

PLOS ONE

Additional Editor Comments (optional):

Reviewers' comments:

Reviewer's Responses to Questions

**Comments to the Author**

1. If the authors have adequately addressed your comments raised in a previous round of review and you feel that this manuscript is now acceptable for publication, you may indicate that here to bypass the “Comments to the Author” section, enter your conflict of interest statement in the “Confidential to Editor” section, and submit your "Accept" recommendation.

Reviewer #1: All comments have been addressed

Reviewer #2: All comments have been addressed

2. Is the manuscript technically sound, and do the data support the conclusions?

Reviewer #1: Partly

Reviewer #2: Yes

3. Has the statistical analysis been performed appropriately and rigorously? 

Reviewer #1: Yes

Reviewer #2: Yes

4. Have the authors made all data underlying the findings in their manuscript fully available?

Reviewer #1: Yes

Reviewer #2: Yes

5. Is the manuscript presented in an intelligible fashion and written in standard English?

Reviewer #1: Yes

Reviewer #2: Yes

6. Review Comments to the Author

Reviewer #1: Please do a throughough final review to correct the few remaining typos and grammar issues in the manuscript (e.g. consistent spacing before parantheses, periods at end of sentences, etc).

Reviewer #2: After revising, the author had done an excellent job addressing prior review comments and concerns. The author had rearranged the description of tables and figures. Besides, the author had added the quantitative findings in the discussion section. Author also unified the description of the screening criteria, which made manuscript more scientific.

Therefore, I recommend this manuscript to be accepted.

7. PLOS authors have the option to publish the peer review history of their article (what does this mean?). If published, this will include your full peer review and any attached files.

Reviewer #1: No

Reviewer #2: No

---

## [Editor Report · Acceptance letter]

26 Feb 2021

PONE-D-20-30406R3 

Poor adherence to TB diagnosis guidelines among under-five children with severe acute malnutrition in central India: a missed window of opportunity? 

Dear Dr. Dhingra:

I'm pleased to inform you that your manuscript has been deemed suitable for publication in PLOS ONE. Congratulations! Your manuscript is now with our production department. 

Kind regards, 

on behalf of

Dr. Frederick Quinn 

Academic Editor

PLOS ONE